# The ectoparasitic seal louse, *Echinophthirius horridus*, relies on a sealed tracheal system and spiracle closing apparatus for underwater respiration

Anika Preuss [1] ✉, Thomas Schwaha[2], Alexander Kovalev [1], David Ebmer[3,4], Insa Herzog [5], Kristina Lehnert[5], Corvin Grass[1], Freya Sandberg[1], Elias Hamann[6], Marcus Zuber[6], Thomas van de Kamp [6,7] & Stanislav N. Gorb [1]

Marine mammals host a diverse array of parasites engaged in a continuous evolutionary arms race. However, our understanding of the biology of parasitic insects associated with marine mammals, particularly their adaptations to challenging marine environments, remains limited. The seal louse, *Echinophthirius horridus*, which infests true seals, is one of thirteen insect species capable of enduring prolonged dives in open seas. This ectoparasite has evolved several adaptations to withstand extreme conditions, such as low oxygen levels (hypoxia), temperature fluctuations, hydrostatic pressure, and strong drag forces during dives. To prevent drowning during their host's 20–35 min dives, seal lice have developed specialized respiratory mechanisms that allow them to survive in oxygen-poor waters and at depths up to 600 m. Advanced imaging techniques, including CLSM, SEM, synchrotron *X*-ray microtomography, and histological sectioning and 3D-reconstruction, have revealed a specialized spiracle closing apparatus for storing oxygen in their tracheal system. Furthermore, our buoyancy experiments showed that the lice consume oxygen under water and, with morphological data, provide what is to our knowledge the first direct evidence against plastron presence. These findings enhance our understanding of the physical adaptations of lice and their survival in extreme ecological conditions, contributing to broader ecological and evolutionary theories.

All conceivable terrestrial habitats have already been conquered by insects, but only 75,000 of 1 million named insect species are able to survive in an aquatic environment[1–4] and even less can be found in the sea[5]. Indeed, only 1400 insect species live in marine habitats like coastal areas, mangroves and salt marshes, while the pelagic zone remains nearly untouched[5,6]. Thereby, the respiratory system of insects, adapted to atmospheric air, is often considered the primary limiting factor for their colonization of marine habitats[7].

In general, the insect respiratory system consists of tracheae, which are tube-like structures with spiral taenidia, that protect them from collapsing[8–10]. These trachea are in direct contact with atmospheric air via spiracular openings, which can be actively opened and closed depending on the respiratory demands[8,11,12].

For breathing underwater, insects have developed various mechanisms: some insects, like the larvae of *Chironomus* and *Polynema*,

---

[1]Department of Functional Morphology and Biomechanics, Zoological Institute, Kiel University, Kiel, Germany. [2]Department of Evolutionary Biology, University of Vienna, Schlachthausgasse 43, Vienna, Austria. [3]Vienna Zoo, Maxingstraße 13b, Vienna, Austria. [4]Institute of Parasitology, Biomedical Research Center Seltersberg, Justus Liebig University Giessen, Schubertstr. 81, Giessen, Germany. [5]Institute for Terrestrial and Aquatic Wildlife Research, University of Veterinary Medicine Hannover, Büsum, Germany. [6]Karlsruhe Institute of Technology, Institute for Photon Science and Synchrotron Radiation, Kaiserstr.12, Karlsruhe, Germany. [7]Karlsruhe Institute of Technology (KIT), Laboratory for Applications of Synchrotron Radiation (LAS), Kaiserstr.12, Karlsruhe, Germany. ✉e-mail: apreuss@zoologie.uni-kiel.de

https://doi.org/10.1038/s42003-025-08285-4                                                                                             **Article**

have a closed tracheal system and gain $O_2$ through cutaneous respiration[13] or by using tracheal gills, like trichopteran larvae or damselfly nymphs[8,14–16]. Other insects, e.g., dipteran larvae, use an elongated abdomen (siphon) that acts as a breathing tube to stay submerged while taking up atmospheric oxygen[13], whereas certain root-piercing insects, such as Ephydridae and Tipulidae, pierce plants to access intracellular air spaces where they can obtain oxygen[17]. Additionally, all insects, possess hemoglobin that can be used to store oxygen more effectively while diving (e.g., in backswimmers or dipteran larvae)[18–22]. However, there are also some insects, such as corixid bugs (*Agraptocorixa eurynome*), that use hydrophobic microstructures to hold and submerge a bubble of air called a compressible gas gill[23]. They consume the stored oxygen, leading to a decrease in oxygen partial pressure in the bubble. Oxygen diffuses into the bubble from the water while nitrogen diffuses out. As the bubble shrinks, the insects must rise to the water surface to replenish the air bubble periodically[8,24]. Another physical gill is the so-called plastron. However, contrary to the compressible gas gill, the plastron is a constant volume structure created through a dense layer of small superhydrophobic microstructures of various shapes, such as buttress-like structures or small cuticle protuberances (microtrichia)[24–27], which are usually bent at the tip and show densities more than 6 million hairs per mm²[13]. These superhydrophobic structures reduce the surface wettability to a minimum, such that the base of the hair and the insect cuticle remains dry and the insect's spiracles are in direct contact with the air film allowing for continuous extraction of oxygen from the water and thereby enabling the insect to remain submerged for extended periods[26,28].

These various mechanisms demonstrate the incredible adaptations that insects have developed to breathe underwater and survive in aquatic environments. However, life underwater in the open ocean presents a particular challenge for aquatic insects with high salinity, exceptionally high hydrostatic pressure, fluctuating temperatures, and hypoxia[29]. In fact, only one single insect lineage containing thirteen species is known for being capable of remaining long time underwater in the open sea: Echinophthiriidae (Phthiraptera: Anoplura; sucking lice)[30,31]. Echinophthiriid lice are obligate ectoparasites that live attached to the fur of semiaquatic mammals and feed on their blood[32,33]. These lice had to adapt to the challenging environment, when their host returned to the sea from land in the Miocene period[34–36].

One representative of the Echinophthiriidae that faces the challenges of surviving in the open sea, is the seal louse *Echinophthirius horridus*, which parasites true seals (Phocidae), like harbor seals (*Phoca vitulina*) and grey seals (*Halichoerus grypus*)[30,31,37]. While being attached to the seal fur, the seal louse has to withstand a hydrostatic pressure of about 5883.96 kPa at 600 m depth for minimum 20–35 min during deep dives of their hosts[38–41]. Nevertheless, previous studies already showed that echinophthiriid lice are able to survive several days while submerged underwater[42–44], but the question about how these lice breathe underwater has been a matter of various speculations, without agreeing on one particular mechanism[29,42–48]. Some have suggested that the lice can breathe underwater with the help of a plastron, which remains stable thanks to the lice's dense scales[24,46,49]. Other studies, such as that by Leonardi and Lazzari (2014), have shown that lice definitely consume oxygen underwater, but whether this works via skin respiration, a plastron or other mechanisms remains unclear, although the tendency in the literature is now more towards the non-existence of such a plastron[42–44].

Considering this unresolved question, this study aims to shed light on (i) the breathing mechanisms, especially on the potential existence of a plastron, of *E. horridus* when submerged underwater by applying buoyancy experiments[50] and (ii) the structural components supporting this breathing mechanism by using modern imaging techniques, such as cryo-scanning electron microscopy (Cryo-SEM)[51,52], confocal laser scanning microscopy (CLSM)[53,54], 3D reconstructions based on histological sectioning and staining[55–57] as well as on synchrotron *X*-ray microtomography[58,59]. Thereby,

we shed new light on the question of how insects can survive in the deep open sea and contribute to the uncovering of a long-standing mystery.

## Methods
### Animals
Adult seal lice (*E. horridus*; Anoplura; Insecta)[45] were collected during necropsies of harbor seals (*P. vitulina*) and grey seals (*H. grypus)*, which were found dead or moribund at the Baltic Sea coast of Kiel-Schilksee (54.418492, 10.177283) in Schleswig-Holstein between April and November 2023. Specimens investigated in the context of this study originated from seals examined as part of monitoring programs within the stranding network of Schleswig-Holstein to investigate their health status[60–62]. Seal lice were stored in a refrigerator at 4–8 °C in single plastic beakers equipped with paper towels moistened with Baltic Sea water. Furthermore, for morphological analysis, seal lice were collected from harbor seals at the Seal Centre Friedrichskoog, the only authorized center for admission, rearing and rehabilitation of abandoned and sick seals in Schleswig Holstein, between June and September 2023. At the Seal Centre, seal lice were exclusively sampled in non-invasive ways using lice combs and forceps during routine medical procedures to avoid stress and unnecessary handling times of seals. Human head lice were collected alive in Göttingen in 2023 and both human head lice and seal lice for morphological examinations were stored in 70% ethanol. Ethical review and approval are not required for this study because all host animals were found dead, died naturally or were euthanized based on welfare grounds and none of the host animals was killed for the purpose of this study. The authors were not involved in host euthanasia as this was done by a third party (certified seal rangers) for external reasons unrelated to this study. We have complied with all relevant ethical regulations for animal use.

### Scanning electron microscopy (SEM)
Adult seal lice ($n = 3$) were scanned in fresh state using cryo-scanning electron microscopy (SEM) by freezing lice in a cryo stage preparation chamber at −140 °C (Gatan ALTO 2500 cryo preparation system, Gatan Inc., Abingdon, UK). Subsequently, frozen samples were sputter-coated with gold-palladium (thickness 6 nm) and observed with a cryo-SEM Hitachi S-4800 (Hitachi High-Technologies Corporation, Tokyo, Japan) in frozen condition at 3 kV accelerating voltage and −120 °C. Obtained images were processed using Adobe Photoshop CS6 (Adobe Photoshop CS, San José, USA) and Affinity Photo (Serif Ltd, Nottingham, UK).

### Confocal laser scanning microscopy
For CLSM analysis, seal lice ($n = 2$) were transferred into glycerine ( ≥99.5%) and lidded with a high precision cover slip (thickness = 0.170 ± 0.005 mm, refractive index = 1.52550 ± 0.00015, Carl Zeiss Microscopy GmbH, Jena, Germany) prior to scanning. Samples' autofluorescence was examined using a CLSM Zeiss LSM 700 based on an upright Zeiss Axio Imager microscope (Carl Zeiss Microscopy GmbH, Jena, Germany). Four solid-state lasers (wavelength 405, 488, 555, and 639 nm) and associated emission filters (BP420–480, LP490, LP560, LP640 nm) were employed. As outlined by Michels & Gorb, the 405 nm excitation and 420-480 nm emission filter visualized less sclerotized cuticle, possibly rich in resilin[54]. Higher sclerotization regions were detected using 488 and 555 nm laser excitations with filters transmitting emission light with wavelength above 490 nm and above 560 nm correspondingly. The 639 nm laser excitation with a 640 nm long-path emission filter captured extended autofluorescence. Projections were processed with ZEN 2008 software (www.zeiss.de/mikroskopie) and Adobe Photoshop CS6 (Adobe Photoshop CS, San José, USA) for qualitative but not quantitative analysis of cuticle composition[54,63–66]. The distinct colors observed in the autofluorescence images correlate with specific material attributes[54]: specifically, reddish autofluorescence denotes highly sclerotized cuticle, with an increased presence of red hues indicating greater degrees of sclerotization. Greenish autofluorescence corresponds to comparatively resilient cuticle with high amount of chitin, while bluish autofluorescence signifies softer, less-sclerotized (often containing resilin) cuticle regions.

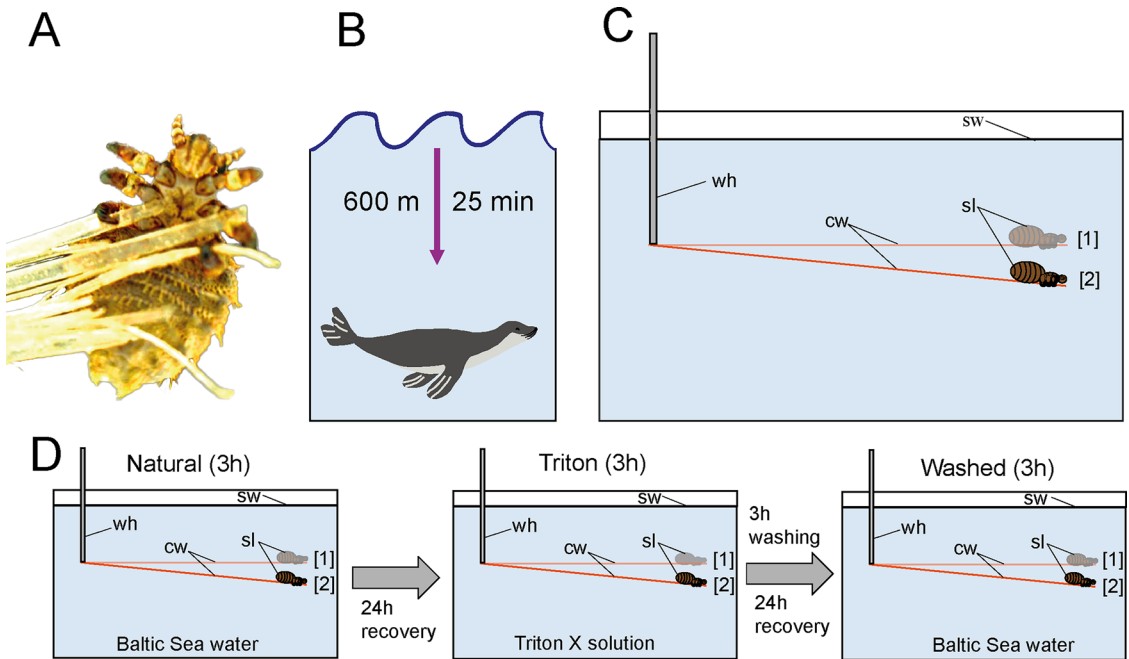

**Fig. 1 | Focal species and experimental setup. A** An adult seal louse, *E. horridus*, on seal fur. **B** Potential diving depth and diving duration of the harbor seal, *P. vitulina*. **C** General experimental setup of the buoyancy experiment with [1] as starting point of the experiment and [2] as final position of the seal louse after 3 h of experiment. **D** Experimental sequence of buoyancy experiment. First, 3 h exposure to Baltic Sea water; after 24 h recovery exposure to Baltic Sea water mixed with 0.1% Triton X followed by 24 h recovery time and 3 h washing process in Baltic Sea water; afterwards, again 3 h exposure to Baltic Sea water. Abbreviations: cw (copper wire); sl (seal louse); sw (sea water); wh (wire holder). Parts of the image were reproduced from Preuss et al. 2024 with the agreement of all authors.

## Buoyancy experiment

In order to evaluate the respiratory behavior of seal lice when submerged underwater, buoyancy experiments were performed in ambient temperature (Fig. 1). To measure the change in volume of the tracheal system and the buoyancy or hydrostatic weighing, respectively, each individual louse (*n* = 13; randomized order) was submerged in a 20 × 14 × 15 cm glass tank while clinging to the end of a 10 cm long piece of Ø 0.1 mm copper wire (Conrad Electronic SE, Hirschau, Germany), imitating a seal hair (Fig. 1). Over a period of 3 h, a picture was taken every minute using a digital reflex camera Nikon D5300 equipped with a Nikon DX SWM VR ED IF Aspherical 72Ø-lense (Nikon Corp., Tokyo, Japan) and a RGBS LCD Timer (RGBS, Zhongshan, China). Each louse went through a sequence of three different treatments on three consecutive days with 24 h recovery time between each experimental step (Fig. 1): (i) for the first 3 h-submersion on the first experimental day, Baltic Sea water (18 a. u., according to PSS-78) was used to simulate the natural habitat of the louse; (ii) on the second experimental day, the louse was exposed to a 0.1% solution of the surfactant Triton X (Sigma-Aldrich Chemie GmbH, Steinheim, Germany) in Baltic Sea water (18 a. u., according to PSS-78) for 3 h. Thereby, Triton X has been shown to act as a wetting agent by reducing the surface tension of water in preceding buoyancy experiments[67]; (iii) in preparation for the third experimental day, lice were washed in Baltic Sea water (18 a. u., according to PSS-78) on a laboratory shaker HS 250 (IKA Werke GmbH & Co. KG, Staufen im Breisgau, Germany), before they were once again submerged in Baltic Sea water (18 a. u., according to PSS-78) for 3 h. For each treatment, separate glass tanks and copper wires were used to prevent skewing results. The three resulting test groups will be referred to as "Natural", "Triton" and "Washed" in the following (Fig. 1).

The first and last picture of the 3 h-test series was analyzed using the software ImageJ (1.53t, Java 1.8.0_322, Wayne Rasband, National Institutes of Health, USA) to detect height shifts of the copper wire over time. For calculating the change in volume of the tracheal system and the buoyancy or hydrostatic weighing, respectively, we calibrated the copper wire with aluminum foil of a known weight and measured height shifts of the wire using ImageJ (1.53t, Java 1.8.0_322, Wayne Rasband, National Institutes of

Health, USA). For the first experimental step, we used the length of the copper wire, 0.1 m, in relation to its length on image in pixels ($n_p$) measured using ImageJ to calibrate pixel size ($c_{pm}$):

$$c_{pm} = \frac{0.1}{n_p}$$

In the next step, we converted the louse height differences in pixels between the first image and the image taken after 3 h (image number 180) ($\Delta y_{[pix]}$) into meters ($\Delta y_{[m]}$) using the pixel size value ($c_{pm}$):

$$\Delta y_{[m]} = \Delta y_{[pix]} * c_{pm}$$

The wire stiffness ($c_w$) was estimated by attachment of the aluminum foil with the mass $m_a$ on it (in air):

$$c_w = \frac{m_a * g}{\Delta y_a},$$

where $g = 9.813 \frac{m}{s^2}$ is the Earth gravitational constant and $\Delta y_a$ is the wire deflection (in meters) with aluminum foil attached. Subsequently, the force $F_{louse}$ responsible for the immersion of the louse in Baltic Sea water after three hours could be calculated using the $c_w$ and $\Delta y_{[m]}$:

$$F_{louse} = c_w * \Delta y_{[m]}$$

Using $F_{louse}$ and the water density value $\rho_{water} = 1022 \frac{kg}{m^3}$ the consumed volume of air $V_{resp}$ could be determined:.

$$V_{resp} = \frac{F_{louse}}{\rho_{water} * g}$$

To exclude the systematic error the general sinking of the wire without a louse attached to it over a period of three hours was also determined for all

three treatments and subtracted from the wire deflection with a louse attached, accordingly. All data are shown in the Supplementary Data 1.

### Synchrotron *X*-ray microtomography and 3D reconstruction

Specimens of *E. horridus* and *P. humanus capitis* ($n = 1$) were scanned in 70% ethanol at the IMAGE beamline of the imaging cluster at KIT light Source. The beam produced by the superconducting wiggler was filtered by 2 mm pyrolytic graphite and monochromatized at 18 keV by a double multilayer monochromator. We employed a fast indirect detector system consisting of a scintillator, visible light optics, a white beam microscope (Optique Peter, Lentilly, France)[68] and a 12-bit pco. dimax high-speed camera (Excelitas PCO GmbH, Kelheim, Germany) with 2016 × 2016 pixels of 11 μm physical size. A magnification of 10x resulted in an effective pixel size of 1.22 μm. For each scan, we took 200 dark field images, 200 flat field images, and 3000 equiangularly spaced radiographic projections in a range of 180° with a frame rate of 50 fps. The control system concert[69] served for automated data acquisition. Data processing including dark and flat field correction and phase retrieval was performed by the UFO framework[70]. The final tomograms were reconstructed with tofu[71] and yielded phase and absorption contrast data sets. These were blended and converted into 8-bit volumes.

Data was segmented and tracheal volumes and surface areas were calculated using Amira 6.2.0 (Thermo Fisher Scientific, Waltham, USA) and rendered with Blender 3.4 (Blender Foundation, Amsterdam, Netherlands) for visualization. Average tracheal diameters of seal lice and head lice were measured based on the CT datasets each 20 times for each category (1st, 2nd, 3rd order trachea and trachea in the head region) for each louse species and the data can be found in Supplementary Data 2.

### Histological sectioning and 3D reconstruction

For histological processing, specimens of *E. horridus* and *P. humanus capitis* ($n = 2$) were dehydrated in a graded ethanol series followed by embedding into Agar LVR resin (Agar Scientific, Stansted, UK) using acetone as intermediate. Serial sections were conducted with a Leica UC6 ultra-microtome (Leica Microsystems, Wetzlar, Germany) at a section thickness of 1 μm. Sections were stained with toluidine blue followed by sealing with a cover slip using the same resin. Sections of spiracles and internal, associated structures were photographed with a Nikon NiU light microscope equipped with a Nikon DsRi2 microscope camera (Nikon Corporation, Tokyo, Japan). Image stacks were aligned and reconstructed with Amira 2021.1.

### Statistics and reproducibility

We compared the descent of all lice under three different treatments ("Natural", "Triton", "Washed") using a one-way analysis of variance with a significance level of 0.05 as the data was normally distributed (criterion for normal distribution: Shapiro Wilk test, $p > 0.05$). All statistical analyses were performed in *R* studio (R version 4.2.1, the R Core Team 2022). R scripts can be found attached in the Supplementary Code.

### Reporting summary

Further information on research design is available in the Nature Portfolio Reporting Summary linked to this article.

## Results
### Morphology of structures potentially involved in breathing under water

Compared to other parasitic lice *E. horridus* has a stocky build with setae covering most of its body (Fig. 2A). These setae exhibit a varying distribution across the body. They are mostly concentrated on the initial segments and middle section of the abdomen, gradually diminishing towards the posterior end and always posteriorly oriented. Two types of setae can be identified based on length: the longer setae extend up to 120 μm, while the shorter ones are less than 25 μm (Fig. 2D). Both long and short setae display similar teardrop-shaped elevation, as illustrated in Fig. 2B–D. In cross-section, both setae types show a kinked shape (45°), although they are anchored in the

cuticle with a straight base (Fig. 2B, C). Shorter setae are distributed throughout the body, whereas longer setae are confined to the abdominal region. Thereby, longer setae typically extend over the recesses of the individual abdominal segments, while smaller setae can be found on the more exposed parts of the abdominal segments. The setae show an auto-fluorescence gradient with yellowish/orange (more sclerotized) to greenish (less sclerotized) from their bases to the tips, indicating a sclerotization gradient. In cross-section, the setae also appear completely filled with the cuticle material without cavities. The density of setae on the abdomen is approximately 400 per square millimeter, which cannot be observed in this density in other parasitic lice such as the terrestrial human head louse (*P. humanus capitis*).

The cuticle surface of *E. horridus* presents a discernible squamous pattern (Fig. 2C), resulting in a body completely covered with plate-like cuticle outgrowths resembling scales due to their preferred orientation. This pattern is interrupted only by single embedded setae. The cross-section and the lateral view show that the individual "scales" do not overlap, but are merely characterized by a constriction of the cuticle between them and a smooth surface (Fig. 2B, C, I). The cuticle is dominated by strong blue autofluorescence and, therefore, presumably, less sclerotized and rather soft. The cuticle material is also not particularly densely packed, but rather characterized by a loose fibrous arrangement (Fig. 2C).

The tracheae of the seal louse have densely packed, bulge-like extensions on the inside, the so-called taenidia, which are separated from each other by small spherical cuticular formations (Fig. 2H). Furthermore, the seal louse has six spiracles on each side of the abdomen (1 per abdominal segment) and each one spiracle on each side of the mesothorax. The spiracles are elevated above the surrounding scales. No setae can be found in the immediate vicinity of the spiracles and no structures directly overhang the spiracles (Fig. 2E, F). A two-part tube-like or flap-like structure is visible within the spiracles, which is characterized on the fluorescence images by a yellowish-orange autofluorescence and thus with an increased degree of sclerotization (Fig. 2F, G). This flap-like structure appears to be part of the cuticular plug as it is connected to the upper part of the plug (cp) (Fig. 3B).

If one compares the aquatic seal louse, *E. horridus*, with the terrestrial human head louse, *P. humanus capitis*, one realizes that both studied lice species have a single, large chamber, the atrium (at), which is the cavity that connects the spiracle (sp) to the tracheal opening. Inside this atrium (at) spinous cuticular processes protrude into the atrial cavity. In *E. horridus* these are fewer and more spaced (Fig. 3A–C) compared to *P. humanus capitis*, where they form a complex dense interconnected, mesh-like network (Fig. 4A, B). In *P. humanus capitis* the atrial cavity (at) is symmetrical in respect to the other cuticular shield (cus). A narrow distal area extends into a broadened area proximally, which is mostly filled by the cuticular mesh, before narrowing down to a proximal tube (att) entering the trachea (Fig. 4C, D). In *E. horridus* the atrium (at) is asymmetrical owing to the two-part cuticular plug (cp) projecting towards the spiracle (sp) opening (Fig. 3E). The atrial cavity (at) is more spacious on the side entering the trachea (tr). Histological sections show a different staining in comparison to the remaining cuticle, which probably reflects more sclerotized cuticle (Fig. 3B). The atrial cavity (at) reaching down to the trachea (tr) is more spacious in comparison to *P. humanus capitis* and the cuticle of the plug (cp) continues along its inner lining as thin, cuticular ridge (atr) (Fig. 3A, B, E, F). On its distal, outer lining the epithelium invaginates into a tube (att), which extends down to the proximal end of the atrial cavity (at) into few terminal branches (Fig. 3D, E). This atrial tube (att) is entirely missing in *P. humanus capitis*.

The transition of atrium (at) towards the trachea (tr) is narrowed. A single cuticular rod (cr) presses against this transition area (Figs. 3B, C, E, F, 4A, C, D). It is elongated and slightly bent in *E. horridus* (Fig. 3C, E, F), whereas much shorter in *P. humanus capitis* (Fig. 4A, C, D). Two large occlusor muscles (ocl) embrace most of the proximal tip of the rod (cr) and emanate to the body wall to attach to the cuticle close to the spiracle (sp) in *E. horridus* (Fig. 3A–C, E, F). In *P. humanus capitis* there are also two, much thinner muscle bundles, one projecting distally towards the spiracle, which

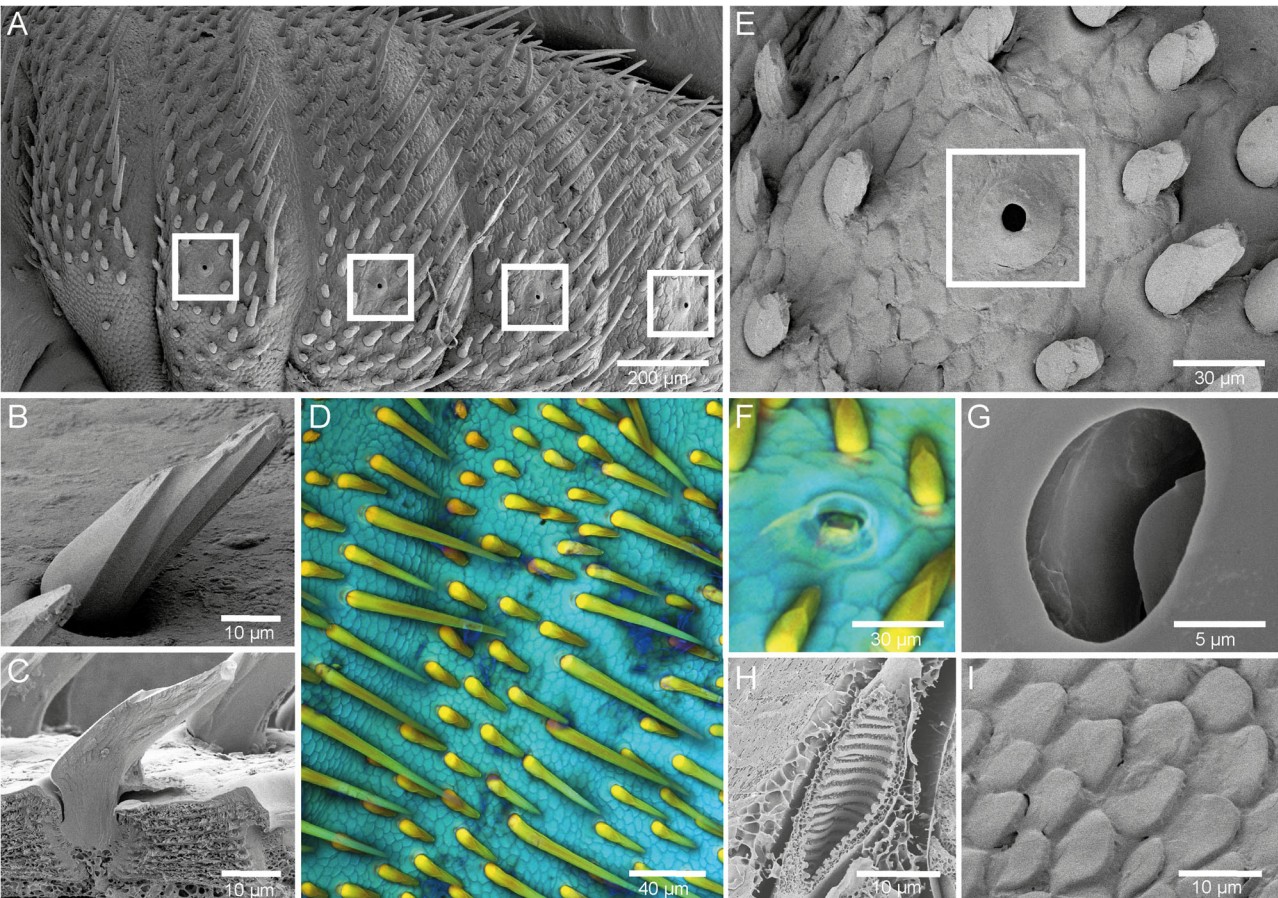

**Fig. 2 | Morphology of the structures potentially involved in underwater breathing of *E. horridus*.** Scanning electron microscopy images of **A** and **E** the abdomen of the specimen from lateral view with spiracles highlighted with white squares, **B** lateral view of a single seta, and **C** a longitudinal section of a single seta and the cuticle. **D** Confocal laser scanning microscopy maximum intensity projection of different setae types from dorsal view. **F** Confocal laser scanning microscopy maximum intensity projection of a spiracle from top view. **G–I** Scanning electron microscopy images of **G** inner structures of a spiracle, **H** an oblique section through a trachea revealing taenidia, and **I** scale-like cuticle outgrowths on the louse surface from dorsal side.

inserts at the lateral border of the spiracle and probably acts as levator of the rod (mrl) and a second bundle running proximally most probably to the outer lining of the trachea, which probably acts as depressor (mrd) (Fig. 4C, D). A third muscle (csd) involved in spiracle closure is present in *P. humanus capitis* that originates from the lateral body wall and inserts at one side of the cuticular shield (cus) where sp is embedded into (Fig. 4). A cuticular valve (cv) is also present at the atrio-tracheal transition in *P. humanus capitis* (Fig. 4A), which is missing in *E. horridus*. The first segment of the tr shows similar spinous cuticular processes as the atrium (at) in *E. horridus* (Fig. 3D), which are not present in *P. humanus capitis*.

### Buoyancy experiment

While being exposed to Baltic Sea water and the surfactant Triton X solution, we were able to observe that the seal lice showed a continuous decrease in buoyancy in all three different treatments over the experimental period of 3 h. No significant differences between the three experimental groups ("Natural", "Triton", and "Washed") could be determined ($p = 0.426$)) (Fig. 5B). On average, we were able to detect a decrease in air volume in the body of the lice in the range of 0.01-0.08 µl for all three different treatments (sd: ±0.015 ("Natural"); ±0.013 ("Triton"); ±0.019 ("Washed")). The progression curves also show that the fastest increase in disappeared air volume in the body of the lice occurred at the beginning, within the first hour of the measurement period (slopes first hour: 0.0004 ("Natural"), 0.0005 ("Triton"), 0.0006 ("Washed")) (Fig. 5A). At the beginning of the third hour, in particular, the decrease in volume was only marginal and the curves flattened out considerably (slopes third hour: 0.0002 ("Natural"), 0.0002

("Triton"), 0.0002 ("Washed")). Thereby, the three treatments differ only insignificantly in their curve progression and show the same trend. Furthermore, we could observe that the lice were still moving on the wire at the beginning of the measurement period in order to find a secure hold, but then they became increasingly stationary as the measurement progressed and finally fell into a state of complete immobility.

### Tracheal system

Both lice species show each one spiracle on each side of the mesothorax and six spiracles on each side of the abdomen (Fig. 6). Thereby, the tracheae differ strongly in diameter from which we derived three categorizations for our study based on our CT data used for 3D-reconstruction: i) Tracheae, which end up in spiracles (first order tracheae); ii) tracheae, which form an "inner ring" around the inner edge of the thorax and abdomen and are highly branched (second order tracheae); and (iii) remaining tracheae, which are next to musculature and inner organs of the lice (third order tracheae) and potentially go over into tracheoles. While the seal louse shows an approximate diameter of about 12.9 µm (sd: ±1.11 µm) for first-order tracheae, the head louse only displays 9.9 µm (sd: ±0.96 µm) for this tracheal order. A similar trend can be seen for the second-order tracheae, which are 12.5 µm (sd: ±1.01 µm) in diameter for the seal louse and 8.3 µm (sd: ±0.95 µm) for the head louse. The third-order tracheae show a diameter of 3.8 µm (sd: ±0.76 µm) in the seal louse and 1.8 µm (sd: ±0.24 µm) in the head louse. Furthermore, the lice' heads are supplied by two smaller sized tracheae (seal louse: 4.8 µm (sd: ±0.92 µm); head louse: 3.0 µm (sd: ±0.73 µm)) and one tracheal tube reaches into each leg of the lice. When

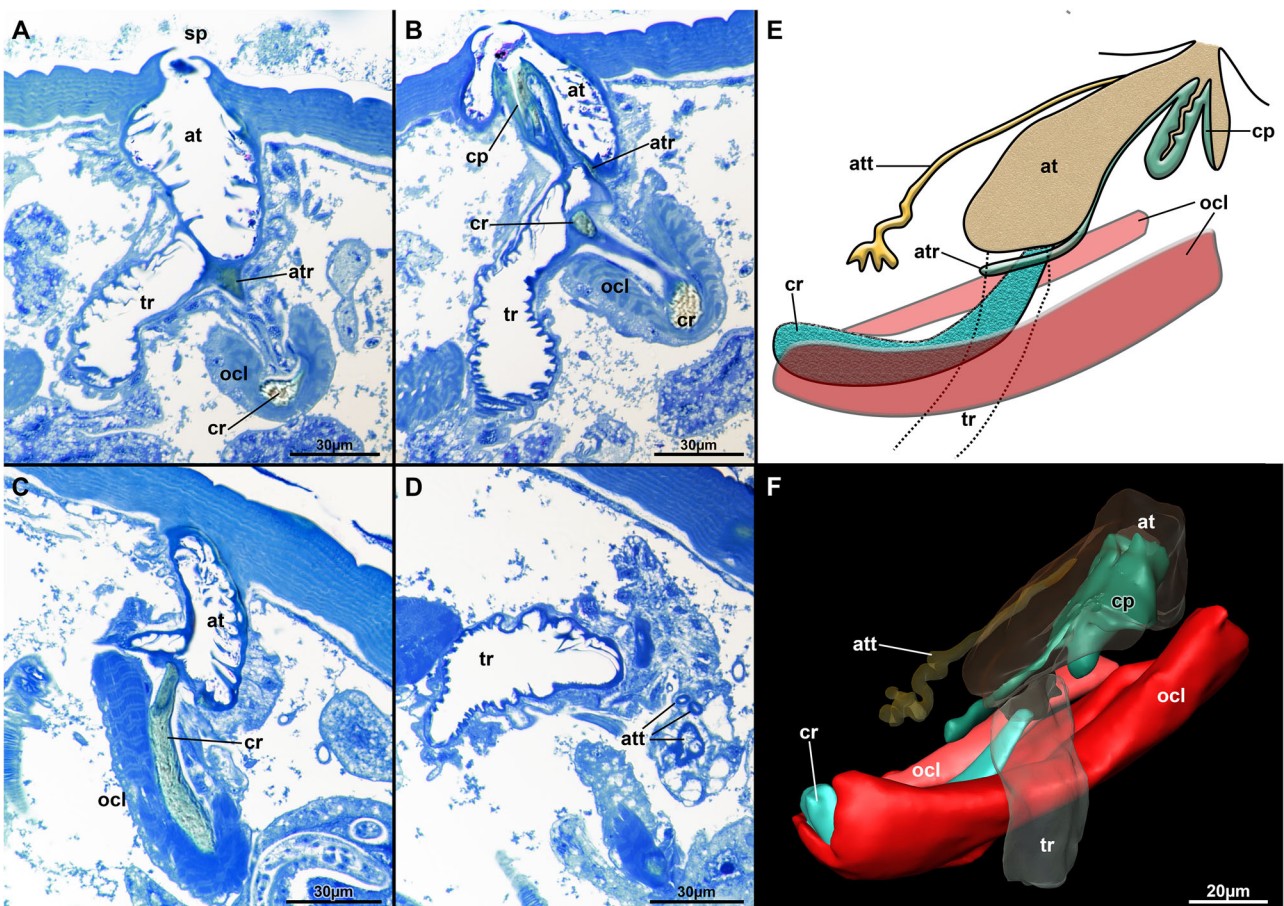

**Fig. 3 | Internal spiracle structure of *E. horridus*. A–D** Histological sections stained with toluidine blue. **A** Spiracle opening with atrium showing spinous processes close to the connection to the trachea. **B** Cuticular plug showing flap-like structure and different staining properties than remaining cuticle. **C** Cuticular rod extending to the atrio-tracheal transition. Also shown are the prominent occlusor muscles. **D** Distal area of the trachea showing spinous processes and also visible the atrial tube.

**E** Schematic drawing of the internal spiracle structures. **F** 3D reconstruction of the internal spiracle structures. Muscles are shown in red and prominent cuticular structures in green/turquoise. Abbreviations: at (atrium), atr (atrial cuticular ridge), att (atrial tube), cp (cuticular plug), cr (cuticular rod), ocl (occlusor muscle), sp (spiracle), tr (trachea).

considering the surface area of the tracheal system in the seal louse (4.902418 mm$^2$) to the whole-body surface (42.142192 mm$^2$), the tracheal system takes up for 11.63%, while in the head louse the area of the tracheal system (3.102167 mm$^2$) in comparison the whole-body surface (28.483300 mm$^2$) makes up 10.89%. Therefore, the tracheal system takes up almost 1.6x as much surface area in the seal louse as in the head louse if one considers the absolute values.

If one considers only the internal volume of the tracheae of the lice (0.005 µl) in relation to the total body volume (2.106 µl) the tracheal system of the seal louse accounts for 0.22% of the total body volume, while the tracheal system (0.003 µl) in the head louse accounts for 0.27% of the whole-body volume (1.309 µl). The values shown for surface area and volume are of course only approximate values, as the very fine tracheoles could not be included due to the resolution limit of the CT scan.

## Discussion

Living on an animal that dives for extended periods, limited oxygen supply and being exposed to the marine environment are major challenges for insects, which is also illustrated by the fact that there are hardly any insects in the open sea[5,6]. Therefore, members of the family Echinophthiriidae, which are lice that parasitize aquatic hosts in the aquatic environment, must possess some exceptional adaptations to be able to survive successfully under water for a longer period of time[30–33]. The seal louse, *E. horridus*, in particular has to face challenging conditions like diving depths of 600 m for 20–35 min by their harbor seal host, *P. vitulina*[39,40]. Furthermore, their host

spends on average only 17% of its time on land as it spends most of its time foraging in the water to catch the 4.6 kg of fish it needs per day[72]. For this reason, the respiratory mechanisms of this louse in particular had to be adapted to life in the marine environment, when its host migrated from land back into the water in the Miocene[34–36,45,46].

Based on our findings, the body of the seal louse, *E. horridus*, is almost completely covered by tear-drop-shaped setae of different lengths (Fig. 2B–D). In previous studies, it was assumed that these setae in combination with the scale-like structures, covering the whole body of the louse, might form a plastron when the insect is submerged under water[24,46]. However, the seal louse shows a hair density of only 400 per square millimeter on its body (Fig. 2D), while other plastron-breathing insects have reported densities of multiple million hair-like cuticle outgrowths per square millimeter[27,73]. Furthermore, the setae of *E. horridus* show a length of 25–120 µm, whereas plastron-bearing structures on other insects usually only reach lengths of 10–20 µm[24]. Therefore, it is unlikely that these setae on *E. horridus* are used to form a plastron. Besides, the spiracle openings are not even covered by these hairs, as is usually the case with a functional plastron, but are rather exposed to the surroundings[24,26,28]. Kim and Ludwig proposed that these setae might have a sensory function, as the lice do not have visual sensory organs[74] and Mehlhorn and colleagues suggest that the lice might collect sebum from their hosts with those hairs for better thermoregulation[49]. However, we think that these hairs have another function that may be related to the drag reduction in these animals: the cross-section of the seal louse setae reminds of the placoid scales of sharks

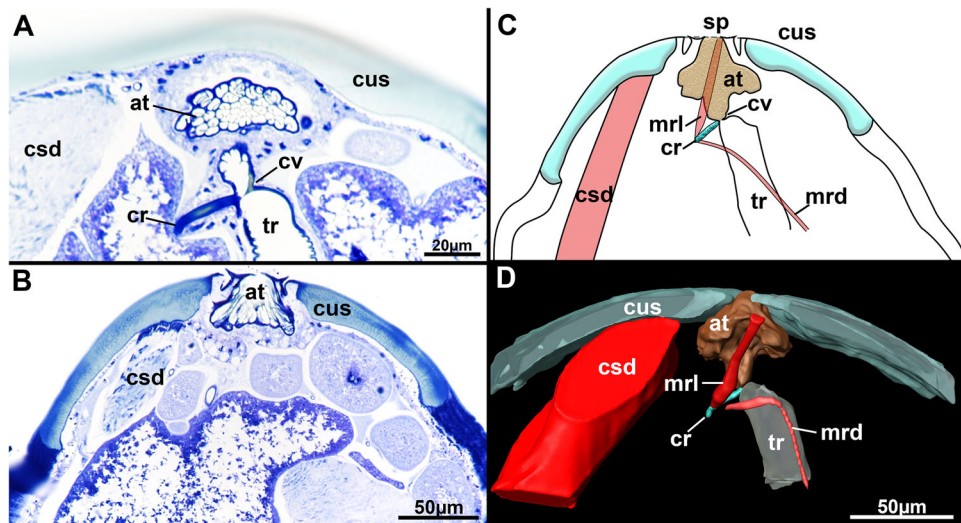

**Fig. 4 | Internal spiracle structure of *P. humanus capitis*. A, B** Histological sections stained with toluidine blue. **A** Transition area of the atrium to the trachea showing the chitinous rod and a cuticular valve. Note also the mesh-like cuticular processes in the atrium. **B** Spiracle with laterally flanking cuticular processes. **C** Schematic drawing of internal spiracle structures. **D** 3D reconstruction of the internal spiracle structures. Muscles are shown in red and prominent cuticular structures in green/turquoise. Abbreviations: at (atrium), cr (cuticular rod), csd (cuticular shield depressor), cus (cuticular shield), cv (cuticular valve), mrd (cuticular rod depressor), mrl (cuticular rod levator), sp (spiracle), tr (trachea).

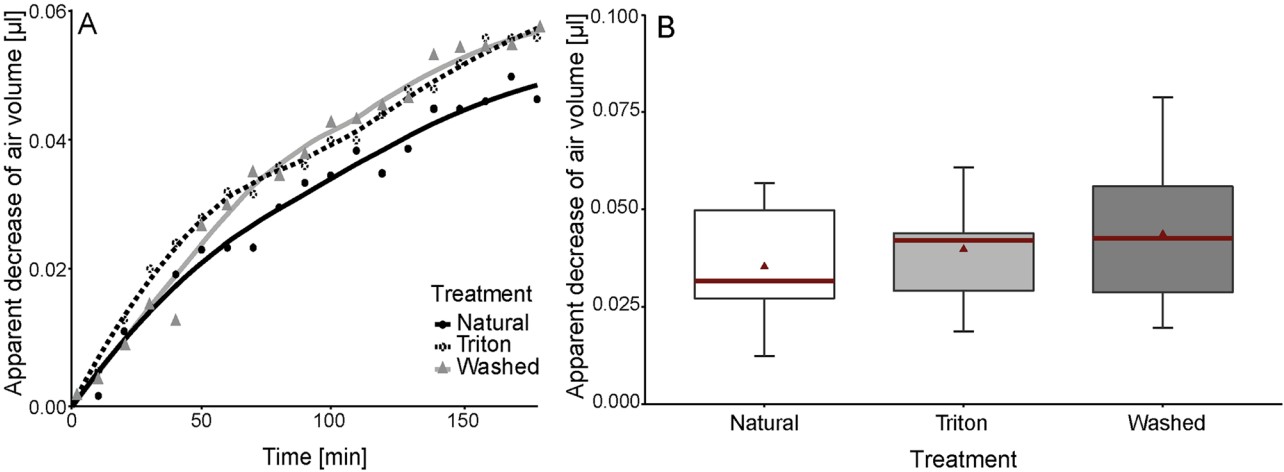

**Fig. 5 | Apparent decrease in air volume calculated from the decrease in buoyant force in the body of *E. horridus* during 3 h submerging period under water with different treatments. A** Exemplary progression curves of the decrease of air volume in the body of one louse in 10 min intervals over a period of 3 h. Black dots and line represent the "natural" condition (exposure to Baltic Sea water), dotted circles and line the exposure to 0.1% Triton X solution in Baltic Sea water, and grey triangles and line symbolize the exposition to Baltic Sea water after contact to Triton X and a 3 h washing period. **B** Total decrease of air volume in μl in the body of the lice (*n* = 13) for the three different treatments ("Natural", "Triton", and "Washed"). Each louse was measured 3 h for each treatment on three consecutive days. The boxes indicate 25 and 75th percentiles, the line within the boxes represents the median, red triangles show the mean values, and whiskers (error bars) define the 10 and 90th percentiles. All data points are plotted.

including ridge-like substructures for drag reduction and better flow characteristics under water[75,76]. This is a new finding that should be investigated in future using a numerical modeling approach.

Additionally, for other Antarctic marine lice species, as *Antarctophthirus ogmorhini* and *Antarctophthirus microchir*, which possess bigger, overlapping scales, it was assumed that these scales might form a plastron[24,49]. The scale-like outgrowths of *E. horridus*, however, are smaller and appear to merge completely into an almost smooth cuticle in the immediate area around the spiracles (Fig. 2I) as also described by Leonardi et al. and Murray et al. for *A. microchir* and *A. ogmohini*[42–44,47]. As a result, the plastron formed by these "scales" would create a layer of air beneath the spiracular opening, which would be isolated from the louse's tracheal system. Thus, a different function of these cuticle structures is more likely. For instance, they might confer flexibility to the cuticle, accommodating the

louse's body as it shrinks and expands with feeding. Also, as suggested by Murray et al., these scales could protect the delicate cuticle of the seal lice and prevent desiccation by trapping water especially when the lice are on land[47]. However, further experiments are necessary to confirm this hypothesis.

Furthermore, we discovered well-developed taenidia inside the trachea of *E. horridus*, which presumably serve to protect the tracheal system from excessive hydrostatic pressure[26,77] and, contrary to the assumption made by Leonardi and colleagues, who assume a possible collapse of the tracheal system during the dives of the seals[38], may prevent the tracheal system from collapsing completely.

We found a very different general structure of the tracheal closing mechanism for *E. horridus* when compared to the observations of Webb[78]. Contrary to the latter, who described a dual spiracular opening, only a single spiracular opening is present in *E. horridus*. Similarly, the internal structure,

**Fig. 6 | 3D-reconstruction of the tracheal system of *E. horridus* and *P. humanus capitis*.** **A** Body of a female *E. horridus* from dorsal view including cuticle (grey and half transparent) and tracheal system (gold). **B** Isolated tracheal system of *E. horridus* (gold) from dorsal view. **C** Body of a female *P. humanus capitis* from dorsal view including cuticle (grey and half transparent) and tracheal system (gold). **D** Isolated tracheal system of *P. humanus capitis* (gold) from dorsal view.

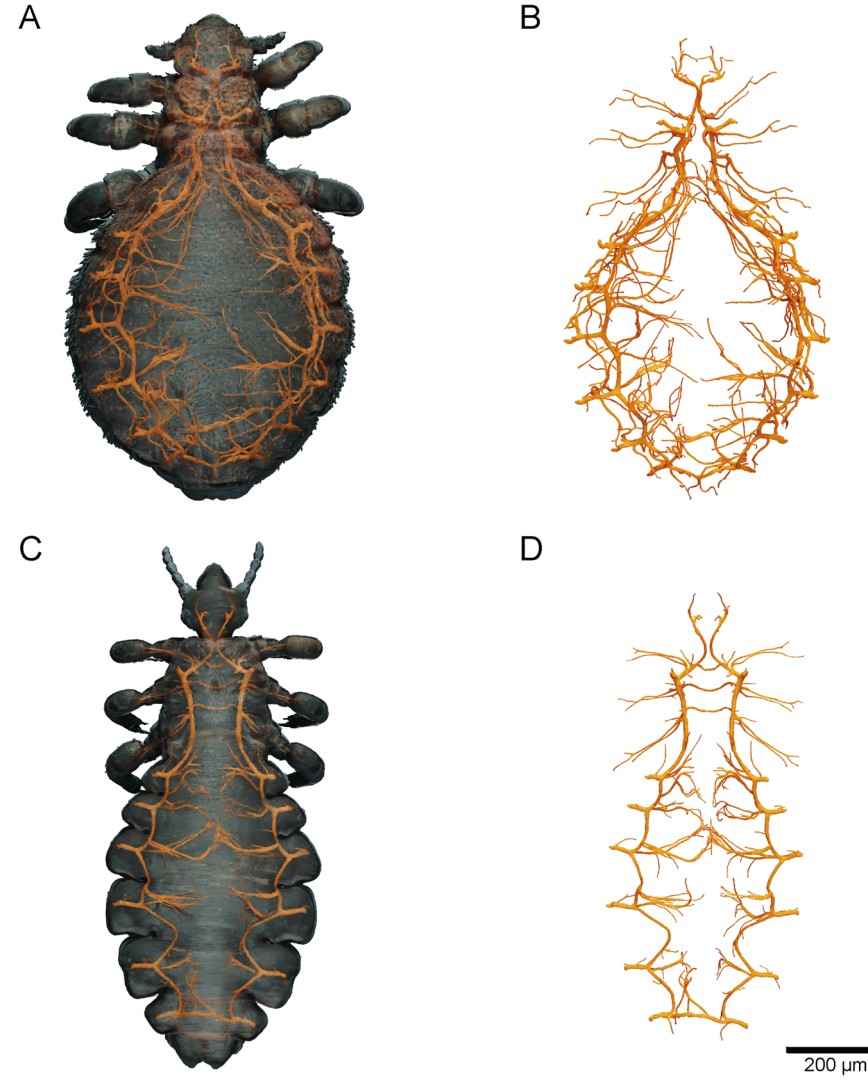

although described in more detail for *A. microchir* as representative of the family, strongly differs from our observations. Whereas Webb described an elongated, distal atrial tube (att) composed of a membranous and sclerotized part[78], we show that *E. horridus* has an asymmetrical atrium (at), mostly membranous with a central cuticular plug (cp). Most probably, the so-called triangular plate of Webb corresponds to our cuticular plug (cp)[78]. As mentioned by Webb, his mounting of the structures may have led to artificial displaced of the different parts[78]. Similarities between his triangular plate and our cp are evident in a thin cuticular connection of the plug to the cuticular rod (cr) and the indication that the knob-shaped triangular plate gets pressed into the distal atrial part by the ocl. The latter were previously only shown by Ass for *E. horridus*, but as a single bundle extending from a comparatively small cuticular rod (cr)[79], contrary to the two different sets of muscles we found in *E. horridus*. Interestingly, the description of Ass features the erroneous dual spiracle opening similar to Webb, but has neither shown a triangular plate nor a cuticular plug (cp)[78,79].

Our Cryo-SEM images reveal that the spiracle opening of *E. horridus* consists of two flap-like structures that presumably form the upper part of the strongly sclerotized cp (Fig. 2F, G; 3B). When the louse comes into contact with water, these flap-like structures close, as the plug is actively pulled inward by muscular action, thereby sealing the spiracle opening and preventing water ingress. Thereby, this structure is reminiscent of the nasal plugs of rorqual whale, where it has been shown that hydrostatic pressure forces these nasal plugs deeper into the nasal passage of those whales, protecting their respiratory tract against water entry and barotrauma[80]. As a

result, only little amounts of water would, if at all, enter the atrium (at) of the respiratory tract of the seal louse since its internal spine structures would potentially hinder water from entering by serving as cage structures since the spines are oriented with their tips into the direction of the spiracle opening. This provides the tracheal system with several protection mechanisms against water penetration: (1) flap-like structures pressed in direction of the spiracle opening based on a muscularly controlled closure system, and (2) internal spines. A novel structure not previously observed is the atrial tube (att) at the distal side of the atrium (at). The narrow lumen does not allow entrance of large quantities of water, but possibly its terminal ends are supplied with glands. Spiracular glands were described for other anoplurans[78], but were not observed so far in *E. horridus* or other echinophthiriids. Better fixations suitable for electron microscopy and transmission electron microscopic observation could clarify the nature and possible function of this atrial tube (att).

Our results on the spiracular closing apparatus of *P. humanus capitis* are more similar to previous descriptions[78] with a large atrium (at) with honeycomb-like structure and a chitinous rod (cr) with musculature for closure. A peritreme, as thin cuticular projection flanking the spiracle opening could be confirmed in our observation. However, we also found numerous differences to Webb's description[78]: a) the entire spiracle (sp) is located on a massive cuticular shield (cus) which itself is pulled inward by a previously unknown, massive cuticular shield depressor (csd). b) our histological investigations first revealed a more sclerotized cuticular valve (cv) at the junction of the atrial to tracheal cavity, opposite of the cuticular

**Fig. 7 | Comparison of tracheal volume to whole body volume in different insect species.** *X*-axis displays drawings of male insect species sorted by their body mass in ascending order. *Y*-axis displays the ratio of the total maximal tracheal volume of the species to their whole-body volume. A detailed list of all reported tracheal volumes and body masses including associated references can be found in the Supplementary Data 3.

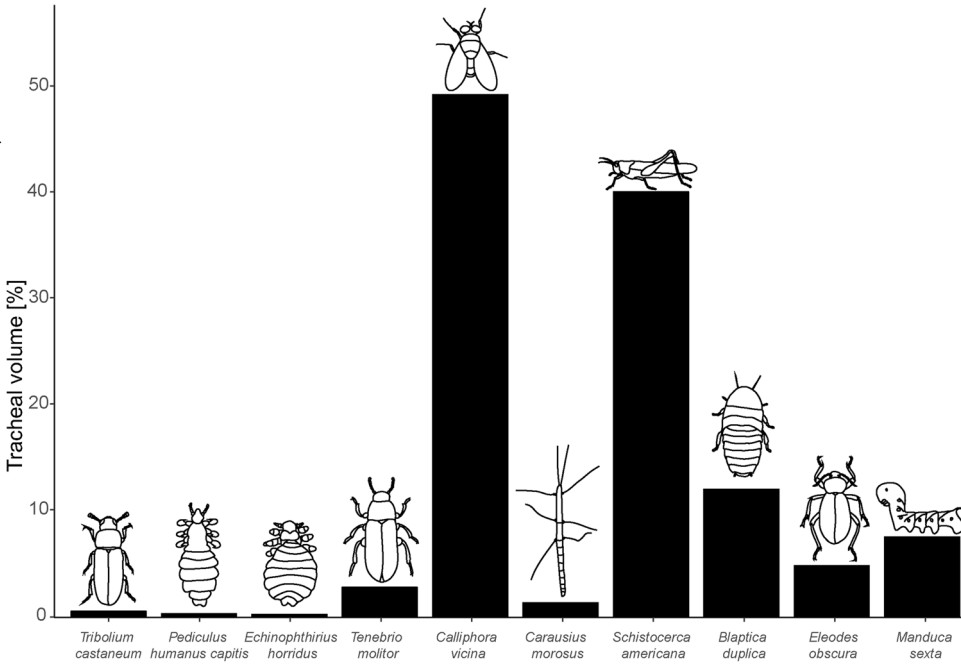

rod (cr) and also two different muscles involved in the movement of the cuticular rod: a levator (mrl) attaching distally to the body wall and a depressor (mrd) most likely originating from the trachea (tr). Since the muscle is very delicate, it was very difficult to trace and requires additional studies and methods to verify this observation. The delicacy of this muscle also explains its absence in Webb's whole mounts[78]. The two different muscles operating on the cuticular rod (cr) most likely buckle the trachea (tr) and thus obstruct the entrance from the atrium (at) into the trachea (tr) at the site of the cuticular valve (cv). On the spiracle side, depression or inward flexion of the cuticular shield (cs) by action of the shield depressor (csd) probably tightens the spiracle opening, along with the surrounding cuticular flaps of the peritreme.

The general structure of spiracle openings in Anoplura is simpler and more similar to the condition of *P. humanus capitis*, whereas the situation found in Echinophthiriidae seems unique with its sophisticated cuticular plug (cp) mechanism, probably as an adaptation for its unique lifestyle, which is why we consider it as an autapomorphy from the spiracle opening structure found in terrestrial Anoplura[78].

In addition to the morphological investigations, we also wanted to experimentally analyze via buoyancy experiments whether the seal lice use a plastron for underwater breathing or not. Therefore, we submerged the lice once in Baltic Sea water as natural condition and once in Baltic Sea water containing 0.1% Triton X, a surfactant known to serve as a wetting agent and thereby eliminating a possible plastron under water[67]. For control, we subsequently washed the lice in Baltic Sea water and submerged them once more under natural condition. In these experiments, we found no difference between the three test groups, as the lice sank on the wire in all three experimental groups (Fig. 5B). Since a plastron is based on the principle of constant volume and gas exchange[24–28,81], if *E. horridus* would rely on plastron respiration, there should not have been any noticeable amount of descent during the natural and control condition, as there would have been a constant respirational gas exchange and an air bubble that would have made them positively buoyant[21]. The Triton X test group on the other hand should have experienced an immediate decrease of air volume right after contact with the Triton X solution as the surfactant would have wet the plastron as shown in Supplementary Fig. 1. There should have been no further descent afterwards and there would have been a high probability of the lice drowning during the three-hour submersion without a plastron. However, we also noticed a further decrease in this experimental group of animals, and the lice

continued to live for days after treatment. This also contradicts the idea that Triton X could have entered the tracheal system through the spiracles, as the lice would otherwise have suffocated within a very short time[23]. Closer microscopic examination did not reveal a plastron in the lice underwater, nor could any air bubbles near the spiracles be identified (Fig. S1)[78,82]. Additionally, a plastron is only stable for diving depths up to 30 m due to the failure of surface tension[48,83,84], which is far surpassed by seals that can dive to depths of up to 600 m[40,41]. We therefore assume, that the seal lice do not possess a plastron, but rather rely on other breathing mechanisms under water.

Furthermore, the progress curves revealed that most air loss happened at the beginning of each experiment with *E. horridus* and slowed down over time (Fig. 5A). During the experiments, we observed that the lice initially moved on the wire on contact with the water, but quickly stopped moving and seemed to fall into a state of complete rigidity. It has been described that the contact with seawater triggers the immobility reflex in several Echinophthiriidae species[38,44]. Thereby, the lices' tolerance to immersion in seawater seems to depend on a reflexive decrease in metabolism and activity, called quiescence, which is triggered by contact with seawater due to high hydrostatic pressure, low temperature, wetting and restricted oxygen availability under water. This quick response helps to conserve energy, nutrients, and oxygen, allowing lice to survive underwater for extended periods[38,44,85,86]. For this reason, the flattening of the curves during our experiments appears to be explainable by the behavioral observations on the lice under water and their associated reduction in metabolism, since this condition enables the lice to ensure their survival without oxygen supply during a long diving period. We also observed that the decrease in air volume was not continuous, but that there were phases in which the decrease was faster. Interestingly, these were always phases in which the lice had moved minimally on the copper wire, presumably to find a better grip, thereby increasing their metabolic rate and oxygen consumption. However, when comparing the decrease of air volume to the actual tracheal volume of the louse it is noticeable that the volume loss is ten times higher than the total tracheal volume (Figs. 5–7). We assume that this observation might be explained by the principle of osmosis: Water molecules diffuse out of the louse's body due to the higher osmotic pressure of the surrounding seawater, leading to water loss. This osmotic water loss reduces the volume of the louse, decreasing the buoyant force acting on it. Simultaneously, the loss of water increases the louse's density, as the density of pure water is lower than

that of seawater and hemolymph. As a result, the louse experiences a net decrease in buoyancy due to the increase in its weight per unit volume[87–90]. Thereby, the lice lose about 0.04 μl body volume per experimental step due to the five-fold higher salt concentration in seawater compared to hemolymph. This effect is also reflected by the progression curves, which show a faster decrease of volume at the beginning of the measurements and a flattening in the later course of the experiments (Fig. 5).

The question now is how these seal lice can actually breathe under water, when the surface plastron is obviously non-existent. In previous studies, it has already been observed that seal lice prefer to stay on the head and flippers of their hosts when they are diving[42,61]. It is assumed that the lice are primarily found in these places, as these are the parts of the body that are most freely stretched out of the water when the seals come up for air. This also gives the louse access to atmospheric oxygen, as the seals can otherwise spend weeks foraging in the open sea without ever coming ashore[61,72]. Accordingly, the lice seem to move to places that are particularly favorable in terms of access to oxygen and feeding, since the head and fins are places that are not advantageous from a fluid mechanics point of view due to strong drag caused by swimming seals[42,91]. There have already been many discussions as to whether the lice possibly obtain all their oxygen requirements through skin respiration by oxygen diffusion through the cuticle, for which a stay on the flippers and the head of the seals could also be advantageous, as the water current and thus the supply of oxygen in these places might be favorable for cuticle respiration[29,42]. However, in the ocean, there is the so-called Oxygen Minimum Zone at a depth of ∼400–800 m, in which the dissolved oxygen in the water is reduced to a minimum[92]. Accordingly, it would be problematic for the lice to be able to absorb oxygen from the water through their skin on the dives of the seals, which can go down to a depth of 600 m[40,41]. During this time, and when staying in anoxic water, the lice have to resort to a different breathing technique. However, in this depths the louse's metabolic demands are also lowest[38,44,85,86] and the seals usually do not spend much time in this region so this assumption should perhaps not be given too much weight. In contrast to fur seals, where the long-haired pelage creates a layer of trapped air[44,86], phocids have a wettable pelage, which prevents the accumulation of air reservoirs, from which the louse could obtain breathing air[24,47,49,93,94]. Therefore, the use of air bubbles trapped in the fur of seals as suggested by Cameron (1956) can be excluded[95].

As our results have now definitively and conclusively shown for the first time experimentally and with the aid of microscopic analyses to the best of our knowledge that seal lice do not possess a plastron, we assume that they rely on respiratory mechanisms already proposed by previous studies: we hypothesize that the seal lice can either store oxygen in pigments like hemoglobin in their body and then release it when needed[29,38,96], use skin respiration[29,42] or store some oxygen in their tracheal system and use this as a kind of additional "storage tank" which becomes smaller as the seals dive deeper. This storage could be always replaced with a new one, when the seals come to the water surface, which could also explain the orientation of the lice on the seal's body[42,61]. If one compares the absolute volumes of the tracheal systems of the terrestrial head louse with those of the aquatic seal louse, it is noticeable that the seal louse has a significantly higher total tracheal volume with considerably more branches and a higher tracheal surface area (Fig. 6A, B). However, if the tracheal volume is set in relation to the body volume, the terrestrial head louse has a larger volume than the aquatic seal louse (Fig. 7). This probably also has something to do with the general body shape of the seal lice, as they generally have a more voluminous abdomen than the head louse, so that the tracheal volume is smaller in relation to the rest of the body volume (Fig. 6). In other insects, the tracheal system often takes up larger parts of the total volume of the body (e.g., *Schistocerca americana* and *Calliphora vicina*), whereby this is due to the larger size (surface-to-volume relationship) or general activity level of the animals (e.g., ability to fly) demanding on oxygen (Fig. 7)[97–99]. Insects that are similarly small in comparison to the two louse species studied, which are only about 2 mm in size, also have a smaller tracheal volume, such as the rice flour beetle, *Tribolium castaneum*, which is able to fly but is only 3–4 mm in size (Fig. 7). Larger insects have to invest more in proportionally lager tracheal

systems than smaller insects, while insects living in hypoxic conditions are known for investing more in the tracheal systems than in hyperoxia conditions[98], also seen in *P. humanus capitis* and *E. horridus*. Other factors that can also influence the tracheal volume are, for example, the presence of air sacs in flying insects, such as flies and grasshoppers[97]. However, since the seal louse is very small, it cannot fly and it can also extremely reduce its metabolism[38,44], it is not dependent on a large tracheal volume and can still live for days on the oxygen in its trachea and the rest of its body.

Furthermore, in the study by Leonardi and colleagues, it was observed that first instar nymphs (N1) of seal lice were able to spend a shorter time underwater without resupplying air without dying[44]. This could be due to the fact that the tracheal system in smaller insects or earlier developmental stages is not yet as well developed as it is the case in adult animals[100,101], also supporting the idea that the tracheal volume and its storage capacity could eventually contribute to the successful underwater respiration of the seal louse. Therefore, we assume that the seal louse uses a mixture of all three contributions for its respiration: (1) skin respiration, (2) pigment respiration and (3) air storage in the trachea. This triple protection enables them to adapt effectively to all kinds of external conditions: breathing in anoxic water, under high pressure or on land. Thereby, the concrete respiratory mechanism of the lice still needs to be investigated for example by carrying out further experiments in the future using respirometry to investigate the influence of pigment and skin respiration on the survival of lice under water.

## Conclusions

The seal louse, *E. horridus*, shows various morphological adaptations for breathing underwater: a highly branched tracheal system, and a special spiracle locking system including flap-like structures at the spiracular opening connected to a plug that is muscularly controlled and internal spines, which protect the rest of the tracheal system from water ingress.

We could not confirm that the scale-like cuticle outgrowths and setae on the louse's body are used for building up a plastron as breathing mechanism as their morphological structure, arrangement and number would not be sufficient to build such a plastron safely under water, nor would this plastron survive at the possible diving depths of seals of up to 600 m. Furthermore, we were unable to confirm the existence of such a plastron air layer via light microscopical imaging or in the course of the diving experiments on the seal lice. Instead, we assume that seal lice use a mixture of skin or pigment respiration and oxygen storage in the tracheal system additionally combined with a strong reduction of their metabolism (quiescence) to survive for days under water.

Based on our results, it would be interesting for future studies to further investigate the breathing mechanisms on a molecular basis and to have a closer look at potential functions of the "scales" and setae on the louse's body. We expect that the Echinophthiriidae have developed further unique adaptations to survive in the marine environment and revealing them and making them usable as possible ideas for technical applications will be a very interesting and challenging task for the future.

## Data availability

All data is provided in the Supplementary Material of the manuscript. Synchrotron data and histological sectioning series[102] can be provided upon request or online under: https://doi.org/10.6084/m9.figshare.28596953.v2.

## Code availability

All scripts used in this manuscript for graphs and analyses are provided in the Supplementary Code of the manuscript.

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

## Acknowledgements

We would like to thank Dr T.B., H.G., F.B., J.T., S.Z., and B.J. for advice and support during this study. Furthermore, we thank E.A. for technical support during the experiments and Prof. O.R. for the provision of Baltic Sea water. We are grateful for the Stranding Network Schleswig-Holstein, and in particular for the seal rangers Sönke Lorenzen, Thomas Diedrichsen and Rolf Lorenzen for alerting us at seal finds. Moreover, we gratefully acknowledge the help of the staff of the Seal Centre Friedrichskoog for the good collaboration and sampling and provision of seal lice. Many thanks also to M.T., who kindly provided us with the head louse samples in ethanol. We thank Dr A.C. for her support during beamtime and Dr T.F. for tomographic reconstructions. We acknowledge the KIT Light Source for provision of instruments at their beamlines and we would like to thank the Institute for Beam Physics and Technology (IBPT) for the operation of the storage ring, the Karlsruhe Research Accelerator (KARA). Funding to S.N.G. by the grant GO 995 46-1 from German Science Foundation (DFG) within the Special Priority Program (SPP 2332) "Physics of Parasitism" is greatly acknowl-edged. The funders took not part in the study design, data collection and analysis, decision of publishing or any preparation of the manuscript.

## Author contributions

A.P. and S.N.G.—conceptualization. A.P., T.S., T.V.D.K., A.K., and S.N.G.—data curation. A.P., T.S., T.V.D.K., A.K., and S.N.G.—formal analysis. K.L. and S.N.G.—funding acquisition. A.P., T.S., T.V.D.K., and A.K.—investigation. A.P., T.S., E.H., M.Z., T.V.D.K., A.K., and S.N.G.—methodology. K.L. and S.N.G.—project administration. K.L. and S.N.G.—resources. A.P. and A.K.—software. S.N.G.—supervision. A.P. and S.N.G.—validation. A.P. and T.S.—visualization. A.P.—writing—original draft. T.S., T.V.D.K., E.H., M.Z., C.G., F.S., I.H., K.L., A.K., D.E., and S.N.G.—writing, review, and editing.

## Funding

## Competing interests

All authors declare no competing interests.
