## [Transparent Peer Review file · Communications Biology]

The ectoparasitic seal louse, *Echinophthirius horridus*, relies on a sealed tracheal system and spiracle closing apparatus for underwater respiration

Corresponding Author: Ms Anika Preuss

Version 0:

Reviewer comments:

Reviewer #1

(Remarks to the Author)

This is a very sound and comprehensive study on the respiratory system of the seal lice.

The excellent anatomical documentation, also on a histological level, combined with the buoyancy experiments shed light both on the structure and function of the respiratory system of the studied insects. The text is well written and gives a good explanation of the conducted experiments and obtained morphological data.

Please find below minor corrections and improvement suggestions.

Line 95: please change "matter" to "a matter"

Lines 217-218: verb is missing in the second part of the sentence "...and Blender 3.4 (Blender Foundation, Amsterdam, Netherlands) for visualization and rendering."

Lines 339-344: This is a very nice comparison of tracheal volumes. I suggest to move it to the discussion section of the text, in order to sustain the clear manuscript structure.

Figures

It is probably a mistake that has occurred during the submission process: I did not find figure legends anywhere, neither in the manuscript file nor in the additional submitted files. Therefore, unfortunately, I could not fully review the figures in the submitted manuscript. I hope this can be easily solved.

Figure 3E: In this drawing, it looks like the hind occlusor muscle is placed in front of the cuticular rode, which is misleading. In case it is a mistake, please correct the drawing, otherwise the schematic drawing does not make a lot of sense.

Although the data availability statement says that "All data is provided in the supplementary material of the manuscript", there is no information on the availability of the synchrotron data or the histological section series images (or for instance where the histological sections are being stored).

Overall, in my opinion this study is a valuable contribution to the study of insect biomechanics. I recommend this manuscript for publication after minor corrections are undertaken. I hope to see the figure legends, since their absence made it more difficult for me to fully review the figures.

Reviewer #2

(Remarks to the Author)

The article „Tracheal system, spiracle closing apparatus, buoyancy, and oxygen consumption in the ectoparasitic seal louse *Echinophthirius horridus*” by Preuss et al. deals with the tracheal system of a ectoparasitic louse.

In general, the article provides several new findings which are quiet interesting.

However, I have some general comments which need improvement from my point of view:

1) The results are very poorly documented. This largely comes from the fact, that the result section is not well structured (and partly repetitive) while other parts are completely missing and the fact that there are neither the abbreviations in the figures

provided nor that the figures have a legend that explain the abbreviations. I would suggest to restructure the results and explain all points raised in the discussion in detail in word and figure (details see below).

The same applies to some measurements. Values for the diameter are provided but is completely unclear if they are averages or how they were measured.

2) The article is not very clear with own and already existing results. For example, it was suggested by previous studies that those lice have no plastron as the spiracles are free of the respective structures. However, this is not clearly mentioned and instead I got the impression that the article sounds as if they found that for the first time. The same applies for example to the different other breathing hypotheses where the current article provides very few to no new results. I would suggest to tone down several sentences like "We suggest that the lice can either store oxygen in pigments and then release it when needed 24, 33 or they can store oxygen in their tracheal system and use this as a kind of "storage tank"" to something like: "as our results further corroborate that seal lice have no plastron, we assume that they rely on respiratory mechanisms already proposed by previous studies".

3) I cannot follow the line of argumentation concerning the air loss during diving. Obviously the measured value cannot be only air (as it is 10 times the volume of the tracheal system). Even when considering that the authors could not reconstruct the largest part of the tracheal systems (all small trachea and tracheaols) this still could not explain that value. However, in the discussion the authors state that the tracheal system serves as air storage during diving. If it is emptied, it cannot store air. So the observed buoyancy effect cannot (at least exclusively) rely on air loss. The authors suggest forward osmosis as a solution which is completely independent from the tracheal system.

4) I do not understand the way, articles are cited in the text. Usually numbers are provided but sometimes also Name and year (without a number) or only name.

5) In the discussion, the authors do not make very well clear which species they talk about in the respective paragraph.

6) I cannot evaluate the figures as I could not find a legend.

Specific comments.

Line 48: please change to 75 000 (with a space) which makes it much easier to read.

Line 55: I am not really happy with only citing Whiggleworth (1953) here. Please cite more general reviews of the tracheal system and its structure (e.g. Noirot & Noirot 1982; Richards 1951 or Whitten 1983).

Line 57: I am even unhappier with the selection of cited papers here. In my understanding it is absolutely mandatory to cite Nikam, T.B., Khole, V.V., 1989. *Insect Spiracular Systems*, first ed. Ellis Horwood Limited, Chichester when talking about spiracular systems.

Line 65: "Very few insects, such as backswimmers or some dipteran larvae even use hemoglobin to store oxygen more effectively while diving". This statement is wrong. All studied groups of insects have hemoglobin (see review in Dittrich & Wipfler 2021).

Line 66: which insects are that?

Line 250: please explain what the colors or the gradient mean.

Line 263: How many spiracles are there and in which segments are they located.

Results: I don't understand the individual chapters in the result section. The first one is called outer morphology but also provides information about the internal construction of the tracheal system (e.g. taenidia, atrium, muscles). Information is provided twice independently (about the spiracles e.g.)

Line 270: I don't like the formulation of the sentence as it creates the impression that it is a particularity of the lice to have an atrium. I would rephrase it to "both studied lice species have a single, large chamber, the atrium (at), that is connected to the tracheal opening".

Line 270ff: the description of the morphology of the spiracle is almost impossible to follow. It needs more details (especially given the space and details in the discussion) and references to the figures are also absolute mandatory. The figures lack any labels and I really don't know which label on the histological sections is what structure in the description.

Line 276: what do you mean with symmetrical? In which axis?

Line 321ff: I strongly disagree with your system to divide the trachea into three categories and the subsequent analyses.

First of all, "tracheoles" is a defined term in the literature based on diameter (less than 1 μm) and the absence or chance in taenidia. In your paper, you define all trachea which are neither attached to spiracles or part of the ring as taenidia. With a diameter of 2.4 and 1.4 μm , they are per definition no tracheoles. I also doubt that they are the place where the actual air exchange takes place. I also have trouble with your definition of "ring tracheae". There should be more than one connection between the spiracles (normally there are three large tracheae between each spiracle). I can also see them in the illustration of the seal louse. How did you distinguish between them.

I would suggest that you provide clear definitions and definitely drop the term tracheoles. Maybe you name them "remaining tracheae".

How are those diameters measured? You provide a single value for the diameter of all three groups. How is this value measured? Is it an average value (if so, where is the raw data and what is the derivation?). I would strongly doubt that all trachea of one of your types have the same value. I assume that this is also not measured from the Ct data (as you have a spatial resolution of 2.44 μm , you cannot measure a diameter of 1.4 μm). Please provide clear information where this values come from and how they are measured. If they derive from a single measurement, either perform a statistically sound analyses with multiple measurements or leave the diameter.

Line 331: how was the surface measured? It is not in the M&M. Please make clear that this is the surface of the measured tracheae (those you can reconstruct). There must be much more, if you would consider all the tiny ones that penetrate every single cell which are however, too small to see in your data.

Line 336: the same applies to the volume. It is only your measured volume.

Line 339: The comparison with other species belongs into the discussion, not the results.

Line 304ff: "On average, we were able to detect a decrease in air volume in the body of the lice of 0.01 to 0.08 μl for all three different treatments". Is a change from 0.01 to 0.08 not an increase instead of a decrease or are the numbers wrong? Or do you mean a decrease from 0.09 to 0.08?

Line 347: I would disagree here. Given the vast amount of aquatic (freshwater) insects, the challenge is not to breath under water but 1) to live on an animal that dives for very long where I have no influence to the access to air (which you outline in the next sentences very nicely) and 2) to live in marine habitats.

Line 375ff. The discussion of the scales in those two lice species seems somehow out of place. Why is it not placed in the discussion of the plastron of the seal louse but in a new paragraph. Murray et al. (1965), Murray (1976) and Leonardi & Lazzari (2014) already argued that there is no plastron in those species. They also showed that the scales have no connection to the spiracles.

Line 383ff: I don't understand that the presence of taenidia has to do with those scales. Taenidia are found in almost all insects. Here it sounds as if the main aim of taenidia is to act against hydrostatic pressure. They make sure that the tracheae do not collapse. What is the assumption made by Leonardi et al.? Why is the article of Leonardi et al. not cited with a number?

L 387ff: The discussion about the differences and similarities between the current description and previous ones is almost impossible to review as the description in the result section is not very detailed and lacks references to the figures and I cannot find legends to the figures which explain what I see and what the abbreviations mean. Which spiracle was studied here?

Line 403. This flap-like mechanism is not mentioned in the results. It would be great to have a drawing of the entire spiracle including the proposed mode of opening and closing. I cannot evaluate its significance or uniqueness without information about it.

Line 416+ 420: Now Webb is cited with a number and a year. Ass in line 398 also has no number.

Line 436: are you talking here about the anopluran ground plan as both *P. humanus* and *E. horridus* are part of Anoplura? If so, how did you derive the ground plan?

Line 439: I would suggest to move this entire chapter to the morphological discussion of the plastron above. It makes sense to me to discuss the plastron once and come both from a morphological and experimental approach to the conclusion that it does not exist, instead of doing this twice independently.

Line 481 ff. I don't understand your explanation with forward osmosis. Why does the density increase when water leaves the animal?

Can you determine which parts of the loss come from the tracheal system and which ones from other body parts? As the trachea have taenidia, they cannot completely collapse, which implies that not even the space of the tracheae can be emptied of air. Does it make sense to speak of air loss in this context if it is obviously much more than air?

Line 508: I don't like the formulation "we suggest" when the hypotheses comes from another source and the current study provides no suggestions towards or against this hypothesis.

L. 509: How can air be stored in the tracheal system if above it is stated that all air in the tracheal system is lost?

Line 511 ff. I don't think that the absolute volume has any implication here. Of course, a larger and more voluminous animal requires more tracheal space. It is surprising that the larger seal louse has comparatively less tracheal space than the terrestrial one.

Line 514ff: I don't understand the logic behind this statement. If the tracheal system would primarily supply the pronounced musculature, why is then not more developed in the thorax and the legs where this musculature sits (similar to the flight musculature of aerial insects which has a much higher tracheal density than the rest of the animal). I don't see a particular tracheal concentration in those body parts of the seal louse and also no difference to the head louse.

Line 519: "This probably also has something to do with the general body shape of the seal lice, as they are generally somewhat broader than the more elongated and narrower head lice (Fig. 6)." This is a vague and somehow not understandable statement. Why should broader animals have more tracheae? You provide the reason for this in the following sentences, i.e. larger insects have comparatively larger tracheal systems.

Line 524: you might want to cite Kaiser et al. 2007 in this respect as it compares the tracheal volume over several species to show that it becomes respectively larger.

Line 532: If I remember that publication correct, N1 (and eggs) cannot survive immersion under water at all, why they have to correlate their reproduction with those of the seals.

Line 538: This was the known status concerning this issue. What does your study provide to this? Pigment respiration is not explained anywhere. The same applies to skin respiration.

Line 543: in which respect is the tracheal system of the seal louse "more sophisticated" than the one of the human louse or any other insect?

Line 552: which types of microscopy were used (and how) to image the plastron?

Reviewer #3

(Remarks to the Author)

I enjoyed reading this paper. How these insects are surviving at depth is an interesting problem. The spiracular morphology you describe is fascinating and the pictures and reconstructions are well done. The experimental work is also interesting and ingenious, but I find I have concerns with the buoyancy results and their interpretation in relation to gas exchange. I've given my thoughts below. I hope you find them constructive!

General comments:

The key experiment you didn't do in this study (but would be well worth trying, and would provide definitive information

regarding the insects' respiratory strategy while diving!) is measuring the seal louse's rate of oxygen uptake, both when they are in air and from the surrounding water when they are submerged. Being so small, a cuticular route is entirely possible/plausible for aquatic gas exchange and could be easily measured using an oxygen sensor in a small, closed volume of water containing one or several lice. This would tell you right away if the seal lice are able to breathe water while submerged. If there is little detectable aquatic gas exchange, then this would indicate that internal stores of O₂ and metabolic suppression are the likely mechanisms that allow them to survive prolonged submersion. Manipulating the temperature of the air and the water would also reveal how much temperature influences/suppresses water could suppress their metabolic rate – I imagine the water temperature in the Baltic is pretty low, especially at depth? During the buoyancy measurement experiments, what was the temperature of the water used in the treatments? Was this measured or controlled? And was it representative of the Baltic ocean, or ambient room temperature? Please include this information.

Using a copper wire as both a simulated seal hair and a sensitive force transducer is very clever! While I applaud the lateral thinking, the significance of the data produced by this method in relation to the insects' respiration are not easy to interpret. In particular the large difference between the small volume of air in the tracheal system measured using CT scans and the large decrease in buoyancy are a little concerning. If there was a decrease in buoyancy due to the consumption of O₂ (and subsequent loss of CO₂ into the water) then its buoyancy should fall by ~20% of the initial air volume in the tracheal system (assuming this is the fraction of air that is O₂). The loss of N₂ would increase this further. However, if the louse's tracheal system was completely rigid, then its volume wouldn't decrease at all and the buoyancy of the louse wouldn't change!

However, you recorded a very large decrease in buoyancy relative to the air volume of the tracheal system. The explanation given on L481-485 regarding the change in buoyancy being attributable to "forward Osmosis" could make sense (I'd refer to this simply as "osmosis" – this isn't an industrial application so "forward" vs "reverse" osmosis isn't relevant – the movement of water down a water potential gradient is well established. As such, the engineering references referring to "forward osmosis" are pretty off topic! I'd delete them). But this then obscures the volume change due to O₂ consumption and adds an additional complication as how the louse survives this additional osmotic challenge would require an explanation, too. As the seal lice are out at sea on their hosts for weeks at a time, an inability to limit cutaneous water loss via osmosis could cause the lice could lose a dangerous quantity of body water – unless they can feed while submerged? Seems unlikely as they wouldn't have access to enough O₂ to digest their blood meal. Anyhow, repeating the buoyancy experiment by placing the lice in fresh water (or better, water that is isosmotic with their hemolymph) would answer the question directly!

Alternatively, if you no longer have access to seal lice, what is the volume of a louse's body (from your 3D scans) and can you estimate what fraction of body water would need to be lost to produce the measured decrease in buoyancy if its hemolymph went from being hypo-osmotic (assuming some typical insect hemolymph osmolarity) to isosmotic with sea water? Is this volume loss reasonable?

There is one additional reason to suspect that osmotic water loss isn't responsible for the change in buoyancy, and that is that each louse was measured 3 times over 3 consecutive days, so were submerged in sea water repeatedly. While there was a 24h recovery time between submergence periods, unless the lice were returned to a live seal to feed on fresh blood during this time (and I assume this wasn't the case?), they wouldn't have had the opportunity to re-hydrate between treatments. But each louse showed the same change in buoyancy in each treatment, despite the fact that the lice in the "triton" and "washed" treatments had endured an additional 3 or 6 hours in sea water, which should have increased the osmolarity of their hemolymph and reduced the rate of osmotic water loss in these subsequent treatments. Does this seem reasonable to you? If you think this is likely, I'd re-examine your conclusions based on these observations.

If the lice are positioning themselves on the head and flippers, this could also be an argument for cuticular gas exchange, as the convection of water past the louse in these more exposed regions would enhance ventilation of their body surface. It is interesting to speculate that this would also be the coolest parts of the seal, which could be relevant for lowering the metabolic rate of the louse by reducing its body temperature. However, without placing thermocouples across a seal's body, this is currently unknowable! And I fully agree with you that the presence of a plastron for aquatic gas exchange is readily rejected due to the depth that these lice descend to and the structure and density of their setae.

L80: Change "Thereby" to "However"

L81: "a particular challenge for aquatic insects, with high salinity..."

L113: "moistened"

L177: "buoyancy"

L183: "in pixels"

L184: "hours"

L187: I assume these manipulations were done in air so buoyancy did not need to be accounted for?

L204-205: What is a superconducting wiggler?!

L333-334: I don't think I follow this – the absolute surface area of the tracheal system, of the seal louse is ~1.6x that of the human head louse, not twice as big. And if you are interested in the capacity of the tracheal system to function as an oxygen store, then volume is the more relevant metric. The greater body surface area of the seal louse is interesting, though. Do you have the masses of the lice to calculate mass specific surface area? If the seal lice breathe cutaneously while submerged then a higher S.A./mass ratio could be expected. Fewer hairs could also enhance cuticular gas exchange by reducing the thickness of any stagnant water boundary layer.

L403-406: The structure of the spiracle is reminiscent of the nasal plugs of a rorqual whale, where hydrostatic pressure forces the plugs deeper into the nasal passage, better sealing them at depth. See Gil, K. et al 2020 Rorqual whale nasal plugs: protecting the respiratory tract against water entry and barotrauma, *J. Exp. Biol.* (2020) 223 (4): jeb219691

L498-500: I don't think the presence of an oxygen minimum zone is a particularly compelling reason to discount cuticular gas exchange in these insects. This only occurs at depth (where temperature is lowest) and so where the louse's metabolic demands are also lowest. This is also likely to be a region where fish are not particularly abundant and a region where seals would spend little time while at sea. Being able to breathe water cutaneously in general would still be a winning strategy for a seal louse, as I don't imagine a louse could rely on being exposed to the air during surfacing in order to breathe (unless they all clustered around the seal's nostrils!). Again, this speculation could be resolved if you could demonstrate/refute cutaneous O₂ uptake in submerged seal lice directly using respirometry.

L508-531: The presence of a respiratory pigment respiration seems very speculative without further evidence to support it (are they able to use the oxygen in the ingested seal blood?! That'd be a neat trick!). The use of the tracheal system as an O₂ storage tank is also speculative as this presupposes that the insect's tracheal system doesn't collapse at these depths. This also seems like something you could measure by direct observation in a small pressure chamber under a microscope, or a modification of your copper wire approach, but in a pressure chamber.

L544: I wouldn't describe the tracheal system as "highly sophisticated". It looks like a pretty standard insect tracheal system.

L546: What do you mean by a "multilayered sluice system"? Can you please explain this?

Version 1:

Reviewer comments:

Reviewer #2

(Remarks to the Author)

This is the second round of review for the article „Tracheal system, spiracle closing apparatus, buoyancy, and oxygen consumption in the ectoparasitic seal louse *Echinophthirius horridus*” by Preuss et al.

The review addresses some problems, but does not solve most of the major concerns I had. In general, the review is not very extensive but rather quick and superficial.

The authors added captions to the figures and references to them. However, the morphological description still remains rather poor (especially when the authors state that the morphological results comprise one of their major contribution) and only a few words or sub-sentences were changed. The used abbreviations in the figures (as as cr or cv) are not used in the text which makes it extremely difficult to correlate the text with the figures. I for example do not understand the flap-like structures. What do they exactly look like and do they operate? Are they part of the plug, as suggested from the legend of figure 3? I would kindly repeat to ask the authors to extend the text of the morphological description.

Additionally all my concerns with the measured values remain. It is not explained in the M&M section, how the diameter was measured. The only thing changed was the spatial resolution of the CT scans. How can you measure a diameter of 2.4 μm or 9.4 μm in a scan with 1.44 μm resolution? Was the trachea 1.6 or 6.4 voxels wide? How many measurements are the base for each of those values? Where is the raw data of the individual measurements?

The problem with the own and already existing results remains. The authors added references and rephrased the mentioned sentence but did not change their general line of argumentation. The authors have morphological and experimental evidence that shows that there is no plastron. However, their morphological line of evidence is based on the fact that the scales do not cover the spiracles. This exact line of argumentation was already provided in previous studies who came to the exactly same conclusion. In my understanding, this needs to be made perfectly clear in the manuscript. Leonardi & Lazzari (2014) showed that lice in hypoxic water survived significantly shorter than in normoxic water which they interpreted as evidence for cuticular respiration. In my understanding, there was already strong evidence that there is no plastron. Please make perfectly clear (both in the introduction and the discussion) what different hypotheses for the respiration of those lice already existed and which evidence or arguments were brought forward for each of them and how your results fit in there.

Concerning the experimental part, I still cannot follow the discussion and interpretations about osmosis and the change of volume in the tracheal system, especially when considering the comments from reviewer #3 about the problems with the osmosis hypothesis. I cannot clearly follow, which hypothesis the author favour or not. For example, the drafted scenario assumes a massive loss of oxygen but also suggests that the tracheal system might serve as a "storage tank" for some oxygen.

The question concerning the ground plan and how two anoplurans can differ from Anoplura still remains.

I consider it as a normal procedure in science that all raw (including histological and CT data) data is made publically available and not provided "upon reasonable request". Students leave university, the permanent staff eventually retires and then the raw data is usually lost.

In summary, I would strongly suggest that the authors provide the following:

I) provide a sound status-quo of the current hypotheses and arguments about the respiration of that group and embed their new data in this framework.

II) That they carefully check their entire manuscript for consistency and clarity both in data and results (e.g. measured values such as diameters, enhance morphological descriptions and explain what the described mechanisms do) but also in hypotheses (e.g. have a consistent line for the problem of the oxygen loss and discuss this in terms of consequences and potential problems).

Reviewer #3

(Remarks to the Author)

Thank you for making the effort to incorporate my previous suggestions. I have only a couple of minor comments relating to the buoyancy measurements and your interpretation and discussion:

The changes in submerged apparent weight which results in the deflection of the copper wire are still being attributed to a decrease in internal gas volume (both in your results L310 onward) and discussion (L476 onward). However, you do not present data that can support this conclusion. What you actually measured is the downward deflection of a wire over time which presumably occurs because of the increase in apparent weight of the attached louse. Why their apparent weight increased over time remains speculation, and you acknowledge in the discussion that a decrease in air volume of the magnitude required for this change isn't possible based on the volume of the tracheal system you measured from your scans. To avoid misleading your readers, I would change the figure legend for fig. 5 and the y axis labels for the graphs in fig. 5 to indicate that this is not a decrease in air volume per se, but rather an "apparent decrease in air volume calculated from the decrease in buoyant force". This is at least accurate!

L307: "the seal lice sank continuously in all three different treatments" should be "the seal lice showed a continuous decrease in buoyancy in all three different treatments"

L495-499: "water molecules diffuse out of the louse's body as the surrounding seawater has a higher osmotic pressure, resulting in water loss in the louse and negative buoyancy. As a result, the body of the louse subsequently has a higher density, as the density of pure water is less than that of seawater and hemolymph, and sinks faster until, after some time, equilibrium is reached and the sinking slows down". Can you please rewrite these two sentences for clarity and to remove references to sinking rate? The decrease in net buoyancy occurs because water loss via osmosis decreases in the volume of the louse (which decrease the buoyant force in N) while also increasing its density (which increases its mass, and therefore weight in N, per unit volume).

Revision

Reviewer #1 (Remarks to the Author):

This is a very sound and comprehensive study on the respiratory system of the seal lice. The excellent anatomical documentation, also on a histological level, combined with the buoyancy experiments shed light both on the structure and function of the respiratory system of the studied insects. The text is well written and gives a good explanation of the conducted experiments and obtained morphological data.

➤ *Thank you for your positive feedback!*

Please find below minor corrections and improvement suggestions.

Line 95: please change “matter” to “a matter”

➤ *Thank you, we have changed this. (l.96)*

Lines 217-218: verb is missing in the second part of the sentence “...and Blender 3.4 (Blender Foundation, Amsterdam, Netherlands) for visualization and rendering.”

➤ *Thank you, we have rephrased the sentence. (l. 217 ff.)*

Lines 339-344: This is a very nice comparison of tracheal volumes. I suggest to move it to the discussion section of the text, in order to sustain the clear manuscript structure.

➤ *Thank you, we have moved it into the discussion part. (l. 543 ff.)*

Figures

It is probably a mistake that has occurred during the submission process: I did not find figure legends anywhere, neither in the manuscript file nor in the additional submitted files. Therefore, unfortunately, I could not fully review the figures in the submitted manuscript. I hope this can be easily solved.

➤ *Thank you for this remark, we now made sure that the legends can be found in the main body of the manuscript. Unfortunately, this was a mistake during the submission process. (l. 848 ff.)*

Figure 3E: In this drawing, it looks like the hind occlusor muscle is placed in front of the cuticular rode, which is misleading. In case it is a mistake, please correct the drawing, otherwise the schematic drawing does not make a lot of sense.

- *We corrected the drawing according to the reviewer's suggestion.*

Although the data availability statement says that “All data is provided in the supplementary material of the manuscript”, there is no information on the availability of the synchrotron data or the histological section series images (or for instance where the histological sections are being stored).

- *Thank you for this remark, we included the sentence “Synchrotron data and histological sectioning series can be provided upon reasonable request.” as the data is still part of ongoing research. (l. 622 f.)*

Overall, in my opinion this study is a valuable contribution to the study of insect biomechanics. I recommend this manuscript for publication after minor corrections are undertaken. I hope to see the figure legends, since their absence made it more difficult for me to fully review the figures.

- *Thank you for this kind revision.*

Reviewer #2 (Remarks to the Author):

The article „Tracheal system, spiracle closing apparatus, buoyancy, and oxygen consumption in the ectoparasitic seal louse *Echinophthirius horridus*” by Preuss et al. deals with the tracheal system of a ectoparasitic louse.

In general, the article provides several new findings which are quiet interesting.

- *Thank you for your remarks, they were very helpful!*

However, I have some general comments which need improvement from my point of view:

1)The results are very poorly documented. This largely comes from the fact, that the result section is not well structured (and partly repetitive) while other parts are completely missing and the fact that there are neither the abbreviations in the figures provided nor that the figures have a legend that explain the abbreviations. I would suggest to restructure the results and explain all points raised in the discussion in detail in word and figure (details see below). The same applies to some measurements. Values for the diameter are provided but is completely unclear if they are averages or how they were measured.

- *Thank you, we now made sure that the labels are integrated in the main body of the text as this was a mistake during the uploading procedure of the submission. We restructured parts of the results and made sure that the reviewer's comments on specific things mentioned in the following are integrated in the manuscript to the best of our knowledge (e.g. diameter measurements). (l. 324 ff.; l. 848 ff.)*

2)The article is not very clear with own and already existing results. For example, it was suggested by previous studies that those lice have no plastron as the spiracles are free of the respective structures. However, this is not clearly mentioned and instead I got the impression that the article sounds as if they found that for the first time. The same applies for example to the different other breathing hypotheses where the current article provides very few to no new results. I would suggest to tone down several sentences like “We suggest that the lice can either store oxygen in pigments and then release it when needed 24, 33 or they can store oxygen in their tracheal system and use this as a kind of "storage tank"” to something like: “as our results further corroborate that seal lice have no plastron, we assume that they rely on respiratory mechanisms already proposed by previous studies”.

- *We have now integrated more references to already existing literature (e.g. for scales). But as it has never been shown experimentally before that seal lice have no plastron when submerged under water, we would like to modify the suggested sentence by the reviewer as otherwise the impression arises that our study makes no contribution to the discussion, although it has now been shown conclusively that no plastron exists. Therefore, we modified the sentence to: “as our results have now definitively and conclusively shown that seal lice do not actually possess a plastron, we assume that they rely on respiratory mechanisms already proposed by previous studies.” Furthermore, we have adjusted our hypotheses in the introduction to make clear that we mainly focused on analyzing the existence of a plastron. (l. 97 ff.; l. 379 ff.; l. 529 ff.)*

3)I cannot follow the line of argumentation concerning the air loss during diving. Obviously the measured value cannot be only air (as it is 10 times the volume of the tracheal system). Even when considering that the authors could not reconstruct the largest part of the tracheal systems (all small trachea and tracheaols) this still could not explain that value. However, in the discussion the authors state that the tracheal system serves as air storage during diving. If it is emptied, it cannot store air. So the observed buoyancy effect cannot (at least exclusively) rely on air loss. The authors suggest forward osmosis as a solution which is completely independent from the tracheal system.

- *Thank you for this remark. We have now somewhat relativized our statement regarding the storage tank and made it clear once again that the focus of the manuscript is not on the respiration methodology itself, but rather on the morphological adaptations and the finding that no plastron is present. (L. 97 ff.; l. 529 ff.)*

4)I do not understand the way, articles are cited in the text. Usually numbers are provided but sometimes also Name and year (without a number) or only name.

- *Thank you for this remark. We have now made sure that we always cited articles via numbers in the text and in the reference list.*

5)In the discussion, the authors do not make very well clear which species they talk about in the respective paragraph.

- *We have integrated the words “seal louse” or “E. horridus” at the beginning of each paragraph. (l. 349 ff.)*

6)I cannot evaluate the figures as I could not find a legend.

- *Thank you for this remark, we now made sure that the legends can be found in the main body of the manuscript. Unfortunately, this was a mistake during the submission process. (l. 848 ff.)*

Specific comments.

Line 48: please change to 75 000 (with a space) which makes it much easier to read.

- *Thank you, we have done this as suggested. (l. 48)*

Line 55: I am not really happy with only citing Wigglesworth (1953) here. Please cite more general reviews of the tracheal system and its structure (e.g. Noirot & Noirot 1982; Richards 1951 or Whitten 1983).

- *Thank you for this input, we have added Noirot and Whitten. (l. 55)*

Line 57: I am even unhappier with the selection of cited papers here. In my understanding it is absolutely mandatory to cite Nikam, T.B., Khole, V.V., 1989. Insect Spiracular Systems, first ed. Ellis Horwood Limited, Chichester when talking about spiracular systems.

- *Thank you, we have changed this. (l. 57)*

Line 65: “Very few insects, such as backswimmers or some dipteran larvae even use hemoglobin to store oxygen more effectively while diving”. This statement is wrong. All studied groups of insects have hemoglobin (see review in Dittrich & Wipfler 2021).

- *Thank you, we have changed this. (l. 64 f.)*

Line 66: which insects are that?

- *Thank you, we have added a specific example including the reference. (l. 66 ff.)*

Line 250: please explain what the colors or the gradient mean.

- *Thank you, we have now explained in brackets what the colors mean. (l. 250 ff.)*

Line 263: How many spiracles are there and in which segments are they located.

- *As we have only mentioned the exact number and location of spiracles in the section “Tracheal system” within the result section, we have additionally included it in “Morphology of structures potentially involved in breathing under water” in the*

result section. (l. 266 ff.)

Results: I don't understand the individual chapters in the result section. The first one is called outer morphology but also provides information about the internal construction of the tracheal system (e.g. taenida, atrium, muscles). Information is provided twice independently (about the spiracles e.g.)

- *Thank you, we have renamed the section in "Morphology of structures potentially involved in breathing under water". Furthermore, we have copied the section about the tracheae and taenidia and included it right after the scale description. The following general description of the spiracles now forms a better transition to the more specialized description about the inner structures of the spiracle system. (l. 239; l. 256 ff.)*

Line 270: I don't like the formulation of the sentence as it creates the impression that it is a particularity of the lice to have an atrium. I would rephrase it to "both studied lice species have a single, large chamber, the atrium (at), that is connected to the tracheal opening".

- *We have changed this. (l. 273 ff.)*

Line 270ff: the description of the morphology of the spiracle is almost impossible to follow. It needs more details (especially given the space and details in the discussion) and references to the figures are also absolute mandatory. The figures lack any labels and I really don't know which label on the histological sections is what structure in the description.

- *We now added more references to the figures in the text, which along with the now added figure legends, which we also slightly extended should allow easier understanding of the morphological structures. (l. 273 ff.; l. 848 ff.)*

Line 276: what do you mean with symmetrical? In which axis?

- *We added more information in the text to refer to its symmetry in regard to the cuticular shield. (l. 278 ff.)*

Line 321ff: I strongly disagree with your system to divide the trachea into three categories and the subsequent analyses.

First of all, "tracheoles" is a defined term in the literature based on diameter (less than 1 μm) and the absence or chance in taenidia. In your paper, you define all trachea which are neither attached to spiracles or part of the ring as taenidia. With a diameter of 2.4 and 1.4 μm , they are per definition no tracheoles. I also doubt that they are the place where the actual air exchange takes place. I also have trouble with your definition of "ring tracheae". There should be more than one connection between the spiracles (normally there are three large tracheae between each spiracle). I can also see them in the illustration of the seal louse. How did you distinguish between them.

I would suggest that you provide clear definitions and definitely drop the term tracheoles.

Maybe you name them “remaining tracheae”.

How are those diameters measured? You provide a single value for the diameter of all three groups. How is this value measured? Is it an average value (if so, where is the raw data and what is the derivation?). I would strongly doubt that all trachea of one of your types have the same value. I assume that this is also not measured from the Ct data (as you have a spatial resolution of 2.44 μm , you cannot measure a diameter of 1.4 μm). Please provide clear information where this values come from and how they are measured. If they derive from a single measurement, either perform a statistically sound analyses with multiple measurements or leave the diameter.

- *Thank you for this advice. We have measured the diameters from our CT data in Amira, but the reviewer is completely right to doubt the correctness due to the mentioned resolution of the scan. As we have actually scanned with 10x magnification at a resolution of 1.22 μm (what we also used for the volume and surface area calculations), we have corrected the resolution values in the M&M section as this was not the right value by mistake. We have now overworked our definitions based on the advice of the reviewer and also integrated clearly that these values are just based on our CT data that can be provided upon reasonable request. Furthermore, we have relativized the section by saying, that we just made these 3 categories for our own study. We set “inner ring” in quotation marks to show that this is just a description of what you can see on our reconstructed image. (l. 210 ff.; l. 324 ff.)*

Line 331: how was the surface measured? It is not in the M&M. Please make clear that this is the surface of the measured tracheae (those you can reconstruct). There must be much more, if you would consider all the tiny ones that penetrate every single cell which are however, too small to see in your data.

- *Thank you for this remark. We have added that we also calculated the tracheal surface with Amira in the M&M section and additionally added a sentence in the result section where we clearly state that the CT has a resolution limit and that we were not able to also calculate the surface area and volume of the very fine trachea reaching into cell areas. (l. 217 ff.; l. 345 ff.)*

Line 336: the same applies to the volume. It is only your measured volume.

- *Thank you, we have added the above-mentioned sentence for more clarity. (l. 345 ff.)*

Line 339: The comparison with other species belongs into the discussion, not the results.

- *Thank you, we have moved it to the discussion. (l. 543 ff.)*

Line 304ff: “On average, we were able to detect a decrease in air volume in the body of the lice of 0.01 to 0.08 μl for all three different treatments”. Is a change from 0.01 to 0.08 not an increase instead of a decrease or are the numbers wrong? Or do you mean a decrease from 0.09 to 0.08?

- *The sentence was indeed worded ambiguously. We have therefore added that it is a decrease in the range of 0.01-0.08 μ l, as this is reflected by the boxplot. (l. 310)*

Line 347: I would disagree here. Given the vast amount of aquatic (freshwater) insects, the challenge is not to breath under water but 1) to live on an animal that dives for very long where I have no influence to the access to air (which you outline in the next sentences very nicely) and 2) to live in marine habitats.

- *Thank you, we have changed this. (l. 350 ff.)*

Line 375ff. The discussion of the scales in those two lice species seems somehow out of place. Why is it not placed in the discussion of the plastron of the seal louse but in a new paragraph. Murray et al. (1965), Murray (1976) and Leonardi & Lazzari (2014) already argued that there is no plastron in those species. They also showed that the scales have no connection to the spiracles.

- *Thank you, we have integrated these references and referred to them. Nevertheless, we would like to keep the discussion about the scales right after the setae as the description and discussion of these structures go hand in hand in our opinion. (l. 379 ff.)*

Line 383ff: I don't understand that the presence of taenidia has to do with those scales. Taenidia are found in almost all insects. Here it sounds as if the main aim of taenidia is to act against hydrostatic pressure. They make sure that the tracheae do not collapse. What is the assumption made by Leonardi et al.? Why is the article of Leonardi et al. not cited with a number?

- *We have integrated the assumption made by Leonardi et al. (2020) in the text and made sure that we cited each manuscript in our study with a number in the reference list and in the text. Furthermore, we have separated the section about taenidia from the section about the function of scales. (l. 391 ff.)*

L 387ff: The discussion about the differences and similarities between the current description and previous ones is almost impossible to review as the description in the result section is not very detailed and lacks references to the figures and I cannot find legends to the figures which explain what I see and what the abbreviations mean. Which spiracle was studied here?

- *Legends have now been added, which should now easily facilitate understanding of the figures. (l. 848 ff.)*

Line 403. This flap-like mechanism is not mentioned in the results. It would be great to have a drawing of the entire spiracle including the proposed mode of opening and closing. I cannot evaluate its significance or uniqueness without information about it.

- *We have now mentioned the flap-like mechanism in the results. We have made sure that the provided drawing of the entire spiracle includes legends and captures for a better understanding of the opening and closing mechanism in addition to the description in the text itself, which we hope is sufficient for a better understanding of*

the entire mechanism. Also, a schematic drawing is present in Fig. 3E. (l. 270; l. 848 ff.)

Line 416+ 420: Now Webb is cited with a number and a year. Ass in line 398 also has no number.

- *Thank you, we have changed this. (l. 415 ff.)*

Line 436: are you talking here about the anopluran ground plan as both *P. humanus* and *E. horridus* are part of Anoplura? If so, how did you derive the ground plan?

- *We have added the number for Webb (1946) after this section to emphasize that we rely his description of the other Anopluran spiracle structures in comparison to *E. horridus* and *P. humanus capitis*. (l. 448 ff.)*

Line 439: I would suggest to move this entire chapter to the morphological discussion of the plastron above. It makes sense to me to discuss the plastron once and come both from a morphological and experimental approach to the conclusion that it does not exist, instead of doing this twice independently.

- *Thank you, this is a very good idea. We have carefully thought about that but found it hard to make a smooth transition from the morphological description of the spiracle closing mechanism to the interpretation of the tracheal volumes if the part about the buoyancy experiments is missing as a link in between. Therefore, we decided to keep the morphological parts together. However, in order to emphasize the difference to morphology, we have now explicitly pointed out at the beginning of this section that the buoyancy experiments are an experimental supplement to the morphological results. We hope, this might make the transition between the parts of the manuscript a little bit smoother. (l. 452 ff.)*

Line 481ff. I don't understand your explanation with forward osmosis. Why does the density increase when water leaves the animal?

Can you determine which parts of the loss come from the tracheal system and which ones from other body parts? As the trachea have taenidia, they cannot completely collapse, which implies that not even the space of the tracheae can be emptied of air. Does it make sense to speak of air loss in this context if it is obviously much more than air?

- *We clarified this statement by explaining that the loss of water in the louse's body results in a higher density of the louse's body as the density of pure water is less than the density of hemolymph and seawater, resulting in negative buoyancy. (l. 496 ff.)*

Line 508: I don't like the formulation "we suggest" when the hypotheses comes from another source and the current study provides no suggestions towards or against this hypothesis.

- *We exchanged "suggest" by "think". (l. 531)*

L. 509: How can air be stored in the tracheal system if above it is stated that all air in the tracheal system is lost?

- *Thank you, we have relativized this part and now stated that they only can store some oxygen in their tracheal system and that this storage becomes smaller when the seal dives deeper. (l. 529 ff.)*

Line 511ff. I don't think that the absolute volume has any implication here. Of course, a larger and more voluminous animal requires more tracheal space. It is surprising that the larger seal louse has comparatively less tracheal space than the terrestrial one.

- *Thank you for this remark. Of course, the absolute volume has no direct implications here. We just mention these values as we have brought them up in the results section and as a transition to the relative volume. To make this clearer, we have deleted the sentence about the pronounced musculature in between as the reviewer is totally right that the reconstructed tracheae are not more pronounced in the legs as in the terrestrial head louse. (l. 536 ff.)*

Line 514ff: I don't understand the logic behind this statement. If the tracheal system would primarily supply the pronounced musculature, why is then not more developed in the thorax and the legs where this musculature sits (similar to the flight musculature of aerial insects which has a much higher tracheal density than the rest of the animal). I don't see a particular tracheal concentration in those body parts of the seal louse and also no difference to the head louse.

- *The reviewer is absolutely right. We have deleted this statement, also making the line of argumentation smoother. (l. 536 ff.)*

Line 519: "This probably also has something to do with the general body shape of the seal lice, as they are generally somewhat broader than the more elongated and narrower head lice (Fig. 6)." This is a vague and somehow not understandable statement. Why should broader animals have more tracheae? You provide the reason for this in the following sentences, i.e. larger insects have comparatively larger tracheal systems.

- *The reviewer is completely right, the statement was confusing. We have now changed this statement by saying that the seal lice generally have a more voluminous abdomen than the head louse, so that the tracheal volume is smaller in relation to the rest of the body volume. (l. 540 ff.)*

Line 524: you might want to cite Kaiser et al. 2007 in this respect as it compares the tracheal volume over several species to show that it becomes respectively larger.

- *Thank you, we added this. (l. 546)*

Line 532: If I remember that publication correct, N1 (and eggs) cannot survive immersion under water at all, why they have to correlate their reproduction with those of the seals.

- *Leonardi et al. state that the median survival (i.e. the time at which 50% of the lice died) for N1 was two days. For eggs, this is completely right, these die in the first 24h of treatment. So at least some of the N1 were able to survive some time under water. (l. 557)*

Line 538: This was the known status concerning this issue. What does your study provide to this? Pigment respiration is not explained anywhere. The same applies to skin respiration.

- *In order to clarify this, we added that skin respiration works by oxygen diffusion through the cuticle and that pigment respiration is based on oxygen storage in pigments like hemoglobin. Furthermore, we made clear that the tracheal storage only slightly contributes to underwater respiration and that the influence of skin and pigment respiration needs to be further investigated by respirometry in the future. (l. 515 f.; l. 529 ff.; l. 566 ff.)*

Line 543: in which respect is the tracheal system of the seal louse “more sophisticated” than the one of the human louse or any other insect?

- *As the reviewer is completely right, we have changed this to “highly branched tracheal system”. (l. 572)*

Line 552: which types of microscopy were used (and how) to image the plastron?

- *Thank you, we have added that we used light microscopy for this (figures provided in the supplements). (l. 578 ff.)*

Reviewer #3 (Remarks to the Author):

I enjoyed reading this paper. How these insects are surviving at depth is an interesting problem. The spiracular morphology you describe is fascinating and the pictures and reconstructions are well done. The experimental work is also interesting and ingenious, but I find I have concerns with the buoyancy results and their interpretation in relation to gas exchange. I've given my thoughts below. I hope you find them constructive!

- *Thank you so much for all these very constructive comments! They helped us a lot.*

General comments:

The key experiment you didn't do in this study (but would be well worth trying, and would provide definitive information regarding the insects' respiratory strategy while diving!) is measuring the seal louse's rate of oxygen uptake, both when they are in air and from the surrounding water when they are submerged. Being so small, a cuticular route is entirely possible/plausible for aquatic gas exchange and could be easily measured using an oxygen

sensor in a small, closed volume of water containing one or several lice. This would tell you right away if the seal lice are able to breathe water while submerged. If there is little detectable aquatic gas exchange, then this would indicate that internal stores of O₂ and metabolic suppression are the likely mechanisms that allow them to survive prolonged submersion.

- *We completely agree with the reviewer and think that respirometry is a great idea for a future study. We have integrated it in the last passage of the discussion and would be happy if the reviewer is maybe willing to cooperate on these experiments with us possibly in the future. In order to make sure that oxygen consumption was not the main important part of the study, we removed this part from the title of the manuscript. (l. 1 ff.; l. 566 ff.)*

Manipulating the temperature of the air and the water would also reveal how much temperature influences/suppresses water could suppress their metabolic rate – I imagine the water temperature in the Baltic is pretty low, especially at depth? During the buoyancy measurement experiments, what was the temperature of the water used in the treatments? Was this measured or controlled? And was it representative of the Baltic ocean, or ambient room temperature? Please include this information.

- *All temperatures during the measurements (water and air) were measured and are stated in detail in the supplementary data. We have included the remark “in ambient temperature” now additionally in the M&M section. (l. 155)*

Using a copper wire as both a simulated seal hair and a sensitive force transducer is very clever! While I applaud the lateral thinking, the significance of the data produced by this method in relation to the insects' respiration are not easy to interpret. In particular the large difference between the small volume of air in the tracheal system measured using CT scans and the large decrease in buoyancy are a little concerning. If there was a decrease in buoyancy due to the consumption of O₂ (and subsequent loss of CO₂ into the water) then its buoyancy should fall by ~20% of the initial air volume in the tracheal system (assuming this is the fraction of air that is O₂). The loss of N₂ would increase this further. However, if the louse's tracheal system was completely rigid, then its volume wouldn't decrease at all and the buoyancy of the louse wouldn't change! However, you recorded a very large decrease in buoyancy relative to the air volume of the tracheal system. The explanation given on L481-485 regarding the change in buoyancy being attributable to “forward Osmosis” could make sense (I'd refer to this simply as “osmosis” – this isn't an industrial application so “forward” vs “reverse” osmosis isn't relevant – the movement of water down a water potential gradient is well established. As such, the engineering references referring to “forward osmosis” are pretty off topic! I'd delete them). But this then obscures the volume change due to O₂ consumption and adds an additional complication as how the louse survives this additional osmotic challenge would require an explanation, too. As the seal lice are out at sea on their hosts for weeks at a time, an inability to limit cutaneous water loss via osmosis could cause the lice could lose a dangerous quantity of body water – unless they can feed while submerged? Seems unlikely as they wouldn't have access to enough O₂ to digest their blood meal. Anyhow, repeating the buoyancy experiment by placing the lice in fresh water (or better, water that is isosmotic with their hemolymph) would answer the question directly!

- *Thank you for these detailed comments. We agree with the reviewer. The body water loss is quite small (see below) and the contraction of soft cuticle, especially in the abdomen, could easily compensate it (see CLSM images). We have changed “forward osmosis” to “osmosis” as suggested. Performing the experiments with an O₂-sensor is a good idea that we will keep in mind for future experiments, thank you.*

Alternatively, if you no longer have access to seal lice, what is the volume of a louse’s body (from your 3D scans) and can you estimate what fraction of body water would need to be lost to produce the measured decrease in buoyancy if its hemolymph went from being hypo-osmotic (assuming some typical insect hemolymph osmolarity) to isosmotic with sea water? Is this volume loss reasonable?

- *If we consider that the density of the louse stays constant, which is a great simplification, as we do not know the change in mass and volume of the louse during the experiments, the body volume change would be 0.04 μ l during each experimental step. This would be a reasonable amount as the total body volume accounts for 2 μ l. But as this is just a simplification, we would not like to implement it in the manuscript file. We just implemented that they lost about 0.04 μ l body volume, what can be also seen from Fig. 5. (l. 496 ff.)*

There is one additional reason to suspect that osmotic water loss isn’t responsible for the change in buoyancy, and that is that each louse was measured 3 times over 3 consecutive days, so were submerged in sea water repeatedly. While there was a 24h recovery time between submergence periods, unless the lice were returned to a live seal to feed on fresh blood during this time (and I assume this wasn’t the case?), they wouldn’t have had the opportunity to re-hydrate between treatments. But each louse showed the same change in buoyancy in each treatment, despite the fact that the lice in the “triton” and “washed” treatments had endured an additional 3 or 6 hours in sea water, which should have increased the osmolarity of their hemolymph and reduced the rate of osmotic water loss in these subsequent treatments. Does this seem reasonable to you? If you think this is likely, I’d re-examine your conclusions based on these observations.

- *As the lice can stay underwater for days, we think that the water loss from their body is getting slower over time also reflected by the progression curves. However, as they were not exposed to freshwater or blood in between the different experimental steps, we think that they continuously lose water over the whole 3 days of treatments but that salt may be excreted by the malpighic vessels during the 24 h resting period in between the experimental steps. (l. 501 ff.)*

If the lice are positioning themselves on the head and flippers, this could also be an argument for cuticular gas exchange, as the convection of water past the louse in these more exposed regions would enhance ventilation of their body surface. It is interesting to speculate that this would also be the coolest parts of the seal, which could be relevant for lowering the metabolic rate of the louse by reducing its body temperature. However, without placing thermocouples across a seal’s body, this is currently unknowable! And I fully agree with you that the presence of a plastron for aquatic gas exchange is readily rejected due to the depth that these lice descend to and the structure and density of their setae.

- *Thank you, we have included the suggestion about the ventilation on the flippers and head. (l. 514 ff.)*

L80: Change “Thereby” to “However”

- *We have changed this. (l. 81)*

L81: “a particular challenge for aquatic insects, with high salinity...”

- *We have changed this. (l. 82)*

L113: “moistened”

- *We have changed this. (l. 114)*

L177: “buoyancy”

- *We have changed this. (l. 178)*

L183: “in pixels”

- *We have changed this. (l. 181)*

L184: “hours”

- *We have changed this. (l. 185)*

L187: I assume these manipulations were done in air so buoyancy did not need to be accounted for?

- *We added “in air” to make this clearer. (l. 189)*

L204-205: What is a superconducting wiggler?!

- *In Synchrotron X-ray Microtomography, a superconducting wiggler is a device consisting of superconducting magnets arranged to create a strong, oscillating magnetic field that forces charged particles into a wavy trajectory, producing high-intensity synchrotron radiation. It is therefore only a component necessary for scanning those samples.*

L333-334: I don’t think I follow this – the absolute surface area of the tracheal system, of the seal louse is ~1.6x that of the human head louse, not twice as big. And if you are interested in the capacity of the tracheal system to function as an oxygen store, then volume is the more relevant metric. The greater body surface area of the seal louse is interesting, though. Do you

have the masses of the lice to calculate mass specific surface area? If the seal lice breathe cutaneously while submerged then a higher S.A./mass ratio could be expected. Fewer hairs could also enhance cuticular gas exchange by reducing the thickness of any stagnant water boundary layer.

- *Thank you, we agree. Unfortunately, the ectoparasite volume is crucially dependent on the feeding status of these animals and the comparison therefore has not much meaning. Indeed, the presence of solid setae enhance the cuticle area and could slightly increase gas exchange. We think that the general shape of the setae supports the building of vortices and reducing the thickness of interface water layer (what should be part of future analyses). We have changed twice as big into 1.6x as big. (l. 340 f.)*

L403-406: The structure of the spiracle is reminiscent of the nasal plugs of a rorqual whale, where hydrostatic pressure forces the plugs deeper into the nasal passage, better sealing them at depth. See Gil, K. et al 2020 Rorqual whale nasal plugs: protecting the respiratory tract against water entry and barotrauma, J. Exp. Biol. (2020) 223 (4): jeb219691

- *Thank you so much, we have integrated this in our manuscript. (l. 417 ff.)*

L498-500: I don't think the presence of an oxygen minimum zone is a particularly compelling reason to discount cuticular gas exchange in these insects. This only occurs at depth (where temperature is lowest) and so where the louse's metabolic demands are also lowest. This is also likely to be a region where fish are not particularly abundant and a region where seals would spend little time while at sea. Being able to breathe water cutaneously in general would still be a winning strategy for a seal louse, as I don't imagine a louse could rely on being exposed to the air during surfacing in order to breathe (unless they all clustered around the seal's nostrils!). Again, this speculation could be resolved if you could demonstrate/refute cutaneous O₂ uptake in submerged seal lice directly using respirometry.

- *Thank you so much, we have included the doubts about the oxygen minimum zone in the manuscript and later on tried to make clear that pigment and skin respiration might be the winning solution. We also integrated that respirometry would be a good idea for future studies. (l. 523 ff.; l. 562 ff.)*

L508-531: The presence of a respiratory pigment respiration seems very speculative without further evidence to support it (are they able to use the oxygen in the ingested seal blood?! That'd be a neat trick!). The use of the tracheal system as an O₂ storage tank is also speculative as this presupposes that the insect's tracheal system doesn't collapse at these depths. This also seems like something you could measure by direct observation in a small pressure chamber under a microscope, or a modification of your copper wire approach, but in a pressure chamber.

- *Thank you, we have integrated this a possible further experiment in the outlook of the discussion. (l. 566 ff.)*

L544: I wouldn't describe the tracheal system as "highly sophisticated". It looks like a pretty standard insect tracheal system.

➤ *We have changed this to “highly branched tracheal system”. (l. 572)*

L546: What do you mean by a “multilayered sluice system”? Can you please explain this?

➤ *As we have not mentioned this kind of multilayered system before, the reviewer is completely right to ask for an explanation for this term. To be more consistent with the rest of the text, we changed it to “internal spines” as described before in the text. (l. 573)*

Reviewers' comments:

Reviewer #2 (Remarks to the Author):

This is the second round of review for the article „Tracheal system, spiracle closing apparatus, buoyancy, and oxygen consumption in the ectoparasitic seal louse *Echinophthirius horridus*” by Preuss et al.

The review addresses some problems, but does not solve most of the major concerns I had. In general, the review is not very extensive but rather quick and superficial.

The authors added captions to the figures and references to them. However, the morphological description still remains rather poor (especially when the authors state that the morphological results comprise one of their major contribution) and only a few words or sub-sentences were changed. The used abbreviations in the figures (as as cr or cv) are not used in the text which makes it extremely difficult to correlate the text with the figures. I for example do not understand the flap-like structures. What do they exactly look like and do they operate? Are they part of the plug, as suggested from the legend of figure 3? I would kindly repeat to ask the authors to extend the text of the morphological description.

- *Thank you for this remark. We have now integrated all the abbreviations mentioned in the figures at a suitable place in the text in brackets (ll. 282 ff.). Furthermore, we integrated the same abbreviations also in the discussion to be more consistent (ll. 408 ff.). In order to better clarify what exactly the flap-like mechanisms are, which are shown in Figure 2F & G with the help of SEM and CLSM images, we have further supplemented the morphological descriptions and also emphasized that these flap-like structures at the spiracle opening are part of the cuticular plug and thereby close by muscle contraction when the louse comes into contact with water. We have now described the exact functional mechanism in detail in the discussion and also clarified once again in the conclusions that these flap-like structures are part of the cuticular plug and are therefore controlled together with it by the muscles (l. 281 f.; ll. 424 ff.). We have ensured, as was the reviewer's wish in the first review round, that all results that are subsequently discussed in the discussion have already been mentioned in the results. If there are any more specific comments from the reviewer, we would be happy to implement them.*

Additionally all my concerns with the measured values remain. It is not explained in the M&M section, how the diameter was measured. The only thing changed was the spatial resolution of the CT scans. How can you measure a diameter of 2.4 μm or 9.4 μm in a scan with 1.44 μm resolution? Was the trachea 1.6 or 6.4 voxels wide? How many measurements are the base for each of those values? Where is the raw data of the individual measurements?

- *To make sure that we actually measured the tracheal diameters at the different sites in a reliable way, we have now added another Excel spreadsheet to the supplements (S3), in which we provide original measurements of the 4 tracheal types (using our nano-CT images of head and seal louse) named in the manuscript. We then took the average of these values and this can now also be found in the manuscript (including sd values) with the note that it is an average value from 20 individual measurements. In the*

previous review, we did not make it clear that the pixel size is 1.22 μm and not the resolution. For this reason, the diameter can be determined approximately, which we have also noted in the manuscript (ll. 339 ff.).

The problem with the own and already existing results remains. The authors added references and rephrased the mentioned sentence but did not change their general line of argumentation. The authors have morphological and experimental evidence that shows that there is no plastron. However, their morphological line of evidence is based on the fact that the scales do not cover the spiracles. This exact line of argumentation was already provided in previous studies who came to the exactly same conclusion. In my understanding, this needs to be made perfectly clear in the manuscript. Leonardi & Lazzari (2014) showed that lice in hypoxic water survived significantly shorter than in normoxic water which they interpreted as evidence for cuticular respiration. In my understanding, there was already strong evidence that there is no plastron. Please make perfectly clear (both in the introduction and the discussion) what different hypotheses for the respiration of those lice already existed and which evidence or arguments were brought forward for each of them and how your results fit in there.

- *In contrast to previous studies, the morphological evidence presented here includes not only the named scales, which the spiracles do not overlap and which have already been described, as noted by the reviewer, but also the setae, detailing their shape and density on the body of the lice. It was also determined that these structures are unsuitable for the plastron function. In addition, we also described and compared in detail the morphological and functional structure of the spiracle opening of a terrestrial and our aquatic seal louse and found that we were able to identify further functional structures that were not yet present in the description by Webb and Ass. In addition, we have provided an additional hypothesis that the setae probably serve to reduce drag under water and compared them with shark scales. Thus, we have reported far more than was previously known in the literature. Although it was already suspected that the lice do not have a plastron, as the reviewer himself says, this was merely an assumption: for the first time we were able to show direct experimental and light microscopic evidence that no plastron actually exists in the seal lice.*
- *Furthermore, although the study by Leonardi and Lazzari (2014) can be interpreted as an initial indication that a plastron is unlikely, the authors did not provide the experiments supporting such a conclusion.*
- *The reviewer states that: “Leonardi & Lazzari (2014) showed that lice in hypoxic water survived significantly shorter than in normoxic water which they interpreted as evidence for cuticular respiration. In my understanding, there was already strong evidence that there is no plastron.” However, this is neither evidence for the absence of plastron nor for the presence of cuticular respiration. As explicitly mentioned in our manuscript, the plastron is also based on gas exchange with the surrounding water: oxygen must diffuse from the water into the plastron to maintain it, while carbon dioxide from the plastron is dissolving into the surrounding water. During cuticular respiration, oxygen is taken up directly from the surrounding water. If we now place a louse in hypoxic water, it does not matter, whether it has a plastron or not: it can neither absorb oxygen directly from the hypoxic water via the cuticle, nor can it survive for a long time with the help of a plastron as oxygen from it will simply diffuse into the surrounding hypoxic water and is additionally used up over time. This is why the lice in the experiment by Leonardi and Lazzari were able to survive significantly*

longer in normoxic water, because there was enough oxygen independent on the respiratory mechanism. Based on their results and interpretation, however, it is not clear whether they breathe via cuticular mechanisms or via a plastron. That study does not provide direct proof of this. And this is precisely the point that our study clearly shows, both experimentally and through microscopic examinations: there is no plastron. This is exactly what we tried to make clear in our manuscript. In the previous round of reviews, we checked once again that what the relevant literature on the plastron debates in the course of the last 80 years is correctly presented:

- “Nevertheless, previous studies already showed that echinophthiriid lice are able to survive several days while submerged underwater⁴²⁻⁴⁴, but the question about how these lice breathe underwater has been a matter of various speculations, without agreeing on one particular mechanism^{29,42-48}.” (ll. 93 ff.)
- “Based on our findings, the body of the seal louse, *E. horridus*, is almost completely covered by tear-drop-shaped setae of different lengths (Fig. 2B-D). In previous studies, it was assumed that these setae in combination with the scale-like structures, covering the whole body of the louse, might form a plastron when the insect is submerged under water^{24,46}” (ll. 373 ff.)
- “Besides, the spiracle openings are not even covered by these hairs, as is usually the case with a functional plastron, but are rather exposed to the surroundings^{24,26,28}. Kim and Ludwig proposed that these setae might have a sensory function, as the lice do not have visual sensory organs⁷³ and Mehlhorn and colleagues suggest that the lice might collect sebum from their hosts with those hairs for better thermoregulation⁷⁴.” (ll. 381 ff.)
- “Additionally, for other Antarctic marine lice species, as *Antarctophthirus ogmorhini* and *Antarctophthirus microchir*, which possess bigger, overlapping scales, it was assumed that these scales might form a plastron^{24,74}. The scale-like outgrowths of *E. horridus*, however, are smaller and appear to merge completely into an almost smooth cuticle in the immediate area around the spiracles (Fig. 2I) as also described by Leonardi et al. and Murray et al. for *A. microchir* and *A. ogmorhini*^{42-44,47}. As a result, the plastron formed by these “scales” would create a layer of air beneath the spiracular opening, which would be isolated from the louse’s tracheal system.” (ll. 391 ff.)
- “Furthermore, in the study by Leonardi and colleagues, it was observed that first instar nymphs (N1) of seal lice were able to spend a shorter time under water without resupplying air without dying⁴⁴. This could be due to the fact that the tracheal system in smaller insects or earlier developmental stages is not yet as well developed as it is the case in adult animals^{100,101}, also supporting the idea that the tracheal volume and its storage capacity could eventually contribute to the successful underwater respiration of the seal louse.” (ll. 570 ff.)
- Now we have additionally clearly stated in the text that there has been no experimental proof or optical evidence for the existence or non-existence of such a plastron to date and also directly addressed the study by Leonardi and Lazzari 2014 in one sentence and explained why further data was needed. In this way, we have now clearly showed the knowledge gap and clarified the relevance of our study:
 - “Some have suggested that the lice can breathe underwater with the help of a plastron, which remains stable thanks to the lice's dense scales^{24,46,49}. Other studies, such as that by Leonardi and Lazzari (2014), have shown that lice definitely consume oxygen underwater, but whether this works via skin respiration, a plastron or other mechanisms remains unclear, although the tendency in the literature is now more towards the non-existence of such a plastron⁴²⁻⁴⁴.” (ll. 96 ff.)

- *“As our results have now definitively and conclusively shown for the first time experimentally and with the aid of microscopic analyses that seal lice do not possess a plastron, we assume that they rely on respiratory mechanisms already proposed by previous studies: we hypothesize that the seal lice can either store oxygen in pigments like hemoglobin in their body and then release it when needed^{29,38,96}, use skin respiration^{29,42} or store some oxygen in their tracheal system and use this as a kind of additional "storage tank" which becomes smaller as the seals dive deeper.” (ll. 541 ff.)*

Concerning the experimental part, I still cannot follow the discussion and interpretations about osmosis and the change of volume in the tracheal system, especially when considering the comments from reviewer #3 about the problems with the osmosis hypothesis. I cannot clearly follow, which hypothesis the author favour or not. For example, the drafted scenario assumes a massive loss of oxygen but also suggests that the tracheal system might serve as a “storage tank” for some oxygen.

- *As stated in the manuscript, our hypothesis was that the lice rely on oxygen storage in their pigments like hemoglobin or on skin respiration. Additionally, we think that they may store some oxygen in the tracheal system serving as a kind of “storage tank”:* *“As our results have now definitively and conclusively shown for the first time experimentally and with the aid of microscopic analyses that seal lice do not possess a plastron, we assume that they rely on respiratory mechanisms already proposed by previous studies: we hypothesize that the seal lice can either store oxygen in pigments like hemoglobin in their body and then release it when needed^{29,38,96}, use skin respiration^{29,42} or store some oxygen in their tracheal system and use this as a kind of additional "storage tank" which becomes smaller as the seals dive deeper. This storage could be always replaced with a new one, when the seals come to the water surface, which could also explain the orientation of the lice on the seal's body^{42,60}.” (ll. 541 ff.)* *Thereby, we tried to make our hypothesis about their breathing mechanism very clear: pigment & skin respiration, but additionally some oxygen might be stored in the tracheal system as well. We cannot clearly state what kind of respiratory mechanism might be the preferred one, as this was not the intent of our study. We show that these lice have a specialized spiracle opening structure in comparison to terrestrial lice species and that they do not have a plastron. The question of the actual breathing mechanism is one we would like to leave open for future studies as stated in the last paragraph of the discussion: “Thereby, the concrete respiratory mechanism of the lice still needs to be investigated for example by carrying out further experiments in the future using respirometry to investigate the influence of pigment and skin respiration on the survival of lice under water.” (ll. 579 ff)*
- *In order to make clear that this is our actual hypothesis, we therefore exchanged “think” by “hypothesize” in line 543 and also added the word “additional” before “storage tank” in line 546.*
- *Concerning the discussion about of osmosis and the line of argumentation, we have now rephrased the section based on the suggestions by reviewer 3 (see below) to make it more understandable (ll. 508-513).*

The question concerning the ground plan and how two anoplurans can differ from Anoplura still remains.

- *Unfortunately, there seems to be a misunderstanding regarding the terminology “ground plan” at this point. We meant the basic structure of this concrete spiracle opening system in Anoplura, which definitely derives from the terrestrial representatives of Anoplura. The spiracle opening structures of the seal louse are features derived from this basic structure and are therefore autapomorphies. We have now also written this specifically in the manuscript in order to avoid further confusion regarding terminology: “The general structure of spiracle openings in Anoplura is simpler and more similar to the condition of *P. humanus capitis*, whereas the situation found in *Echinophthiriidae* seems unique with its sophisticated cuticular plug mechanism, probably as an adaptation for its unique lifestyle, which is why we consider it as an autapomorphy from the spiracle opening structure found in terrestrial Anoplura⁷⁸.” (ll. 460 ff.)*

I consider it as a normal procedure in science that all raw (including histological and CT data) data is made publically available and not provided “upon reasonable request”. Students leave university, the permanent staff eventually retires and then the raw data is usually lost.

- *At Kiel University, the statutes request that all data is stored on the university's servers when you leave the university so that the data is never lost. Since the data, as already noted in the first review round, are still part of ongoing studies, we would be reluctant to make them available online before the other studies have at least been submitted, which is why it is actually rather common procedure to only make them available on request. However, as we are of course willing to make the data publicly available and have done so with all our other data, we have now uploaded the CT dataset and the histological data to Figshare and included the corresponding DOI in the manuscript. This dataset will be immediately publicly released online with the publication of the manuscript (l. 637).*

In summary, I would strongly suggest that the authors provide the following:

I) provide a sound status-quo of the current hypotheses and arguments about the respiration of that group and embed their new data in this framework.

- *Thank you, we have now made sure that our study is well supported by the previous studies and is embedded in their framework.*

II) That they carefully check their entire manuscript for consistency and clarity both in data and results (e.g. measured values such as diameters, enhance morphological descriptions and explain what the described mechanisms do) but also in hypotheses (e.g. have a consistent line for the problem of the oxygen loss and discuss this in terms of consequences and potential problems).

- *Thank you, we have made sure that our manuscript is consistent regarding data, results, and hypotheses as explained in detail above.*

Reviewer #3 (Remarks to the Author):

Thank you for making the effort to incorporate my previous suggestions. I have only a couple of minor comments relating to the buoyancy measurements and your interpretation and discussion:

The changes in submerged apparent weight which results in the deflection of the copper wire are still being attributed to a decrease in internal gas volume (both in your results L310 onward) and discussion (L476 onward). However, you do not present data that can support this conclusion. What you actually measured is the downward deflection of a wire over time which presumably occurs because of the increase in apparent weight of the attached louse. Why their apparent weight increased over time remains speculation, and you acknowledge in the discussion that a decrease in air volume of the magnitude required for this change isn't possible based on the volume of the tracheal system you measured from your scans. To avoid misleading your readers, I would change the figure legend for fig. 5 and the y axis labels for the graphs in fig. 5 to indicate that this is not a decrease in air volume per se, but rather an "apparent decrease in air volume calculated from the decrease in buoyant force". This is at least accurate!

- *Thank you, we have now changed the y-axis labels in Figure 5 to "Apparent decrease of air volume [μl]" and also adjusted the figure legend by putting the limitation "apparent decrease in air volume calculated from the decrease in buoyant force" in the legend as suggested by the reviewer (l. 897).*

L307: "the seal lice sank continuously in all three different treatments" should be "the seal lice showed a continuous decrease in buoyancy in all three different treatments"

- *Thank you, we have changed this (l. 319).*

L495-499: "water molecules diffuse out of the louse's body as the surrounding seawater has a higher osmotic pressure, resulting in water loss in the louse and negative buoyancy. As a result, the body of the louse subsequently has a higher density, as the density of pure water is less than that of seawater and hemolymph, and sinks faster until, after some time, equilibrium is reached and the sinking slows down". Can you please rewrite these two sentences for clarity and to remove references to sinking rate? The decrease in net buoyancy occurs because water loss via osmosis decreases in the volume of the louse (which decrease the buoyant force in N) while also increasing its density (which increases its mass, and therefore weight in N, per unit volume).

- *We have changed the sentences now to: "Water molecules diffuse out of the louse's body due to the higher osmotic pressure of the surrounding seawater, leading to water loss. This osmotic water loss reduces the volume of the louse, decreasing the buoyant force acting on it. Simultaneously, the loss of water increases the louse's density, as the density of pure water is lower than that of seawater and hemolymph. As a result, the louse experiences a net decrease in buoyancy due to the increase in its weight per*

unit volume." Thereby, we removed references to the sinking rate and included the suggestions made by the reviewer (l. 508-513).